# Generalizable Process Reward Models via Formally Verified Training Data

## Abstract

Process Reward Models (PRMs), which provide step-level feedback on reasoning traces generated by Large Language Models (LLMs), are receiving increasing attention. However, two key research gaps remain: creating PRM training data requires costly human annotation to label accurate step-level errors, and existing PRMs are limited to math reasoning domains. In response to these gaps, this paper aims to enable automatic synthesis of accurate PRM training data and the generalization of PRMs to diverse reasoning tasks beyond math reasoning. We propose FoVer, an approach to synthesize PRM training data with accurate step-level error labels automatically annotated by formal verification tools, such as Z3 and Isabelle. To show the practical effectiveness of FoVer, we synthesize a training dataset by annotating step-level error labels on LLM responses to formal logic and theorem proving tasks, without relying on human annotation. While FoVer creates training data with symbolic tasks compatible with formal verification, our experiments show that PRMs trained on our dataset exhibit cross-task generalization, enabling a single PRM to effectively perform verification across diverse reasoning tasks. Specifically, LLM-based PRMs trained with FoVer significantly outperform PRMs based on the original LLMs and achieve competitive or superior results compared to state-of-the-art PRMs, as measured by step-level verification on ProcessBench and Best-of-K performance across 12 reasoning benchmarks, including MATH, AIME, ANLI, MMLU, and BBH. The dataset and code are in the supplementary material and will be made public.

## 1 Introduction

Process Reward Models (PRMs), which provide step-level feedback on reasoning traces generated by Large Language Models (LLMs), have been increasingly used to improve LLMs by providing

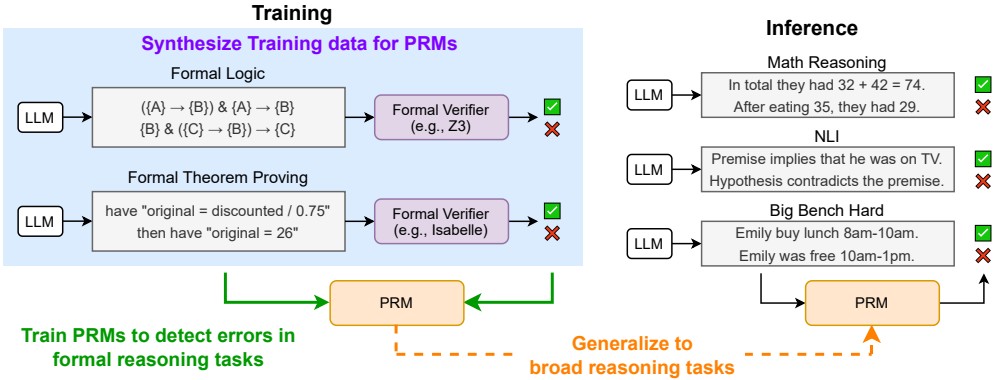

Figure 1: FoVer Overview. Left: We create PRM training data using accurate step-level error labels automatically annotated by formal verifiers. Right: We use synthesized data to train PRMs, and the resulting PRM can be applied to broad reasoning tasks.

Table 1: Comparison of FOVER with prior PRM research. Earlier work relies on costly human annotations or noisy synthetic labels to create training data and is largely limited to math reasoning. In contrast, FOVER leverages formal verification to automatically generate accurate error labels, producing synthetic training data that improves PRM performance across diverse reasoning tasks.

| Research on PRMs | Training Data Creation | | Training Domain | | | Evaluation Domain | | | |
|---|---|---|---|---|---|---|---|---|---|
| | Automatic | Accurate | Math (informal) | Formal Logic | Formal Proving | Math (informal) | Academic Exams | Logic Reasoning | BBH |
| PRM800K (2024) | – | ✓ | ✓ | – | – | ✓ | ✓ | – | – |
| Math-Shepherd (2024a) | ✓ | – | ✓ | – | – | ✓ | – | – | – |
| Qwen2.5-Math-PRM (2025) | ✓ | – | ✓ | – | – | ✓ | ✓ | – | – |
| FOVER (Ours) | ✓ | ✓ | – | ✓ | ✓ | ✓ | ✓ | ✓ | ✓ |

supervision during training and inference (Uesato et al., 2022; Lightman et al., 2024). PRMs are typically created by fine-tuning LLMs on the task of classifying whether each step in LLM-generated reasoning is correct, using training data annotated with step-level error labels (correct or incorrect) on reasoning traces (Wang et al., 2024a; Zhang et al., 2025).

However, there are two major research gaps as shown in Table 1. First, prior work employs costly human annotation (Uesato et al., 2022; Lightman et al., 2024) or noisy step-level error labels (Wang et al., 2024a; Luo et al., 2024) to create training data for PRMs. Second, PRMs have often been studied only on mathematical reasoning, and thus the verification capabilities of PRMs outside of math reasoning are not well studied (Zeng et al., 2025; Yin et al., 2025). To address these gaps, this work advances the state-of-the-art of PRMs in two directions: (1) automatic, accurate creation of training data for PRMs, and (2) generalization of PRMs to diverse reasoning tasks.

We propose FOVER (Figure 1), a novel approach to create PRM training data using formal verification tools, such as Z3 (de Moura & Bjørner, 2008) and Isabelle (Nipkow et al., 2002). FOVER creates PRM training data by using these tools to provide accurate step-level error labels on LLM responses to symbolic logic tasks like formal theorem proving, without requiring human annotation.

To show the practical effectiveness of FOVER, we create PRM training data by annotating step-level error labels on LLM responses to formal logic and theorem proving tasks using formal verification tools, which we refer to as FOVER-80K. Then, we fine-tune Llama 3.1 8B (Llama Team, 2024) and Qwen 2.5 7B (Qwen Team, 2024) on FOVER-80K to use them as PRMs. We evaluate these PRMs in two settings, Best-of-K (Cobbe et al., 2021; Li et al., 2023) on diverse reasoning tasks (§4.2) and step-level verification on ProcessBench (Zheng et al., 2025) (§4.3), both of which are standard practices for assessing PRMs (Lightman et al., 2024; Wang et al., 2024a).

Our experiments show that the LLM-based PRMs fine-tuned on FOVER-80K significantly outperform baseline PRMs based on the original LLMs and are competitive with or better than state-of-the-art PRMs built on the same LLMs (Xiong et al., 2024; Zheng et al., 2025; Zhang et al., 2025). Surprisingly, our experiments on 12 reasoning benchmarks show that training on FOVER-80K—comprising of error labels on formal logic and theorem proving tasks—exhibits cross-task generalization and enhances Best-of-K performance of PRMs across diverse out-of-distribution reasoning tasks, including MATH (Hendrycks et al., 2021), AIME (Art of Problem Solving, 2025), ANLI (Nie et al., 2020), MMLU-PRO (Wang et al., 2024b), and BBH (Suzgun et al., 2023) (Figure 2).

Our main contributions are as follows:

- We propose FOVER, a novel method to generate PRM training data with step-level error labels annotated by formal verification. FOVER is the first approach to synthesize PRM training data that is both automatic and accurate.

- We create FOVER-80K, a novel training dataset for PRMs consisting of reasoning traces generated from Llama 3.1 and Qwen 2.5 for formal logic and theorem proving tasks, with step-level error labels annotated by Z3 and Isabelle.

- While FOVER creates PRM training data with formal reasoning tasks, we demonstrate that training on FOVER-80K improves PRMs on broad out-of-distribution reasoning tasks, showing the practical effectiveness of FOVER. Our PRMs perform competitively with state-of-the-art PRMs, as measured in 12 reasoning benchmarks.

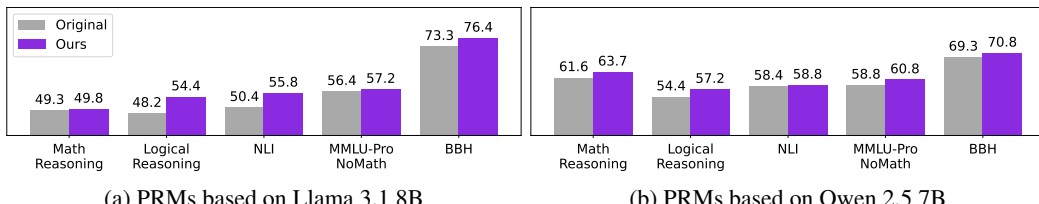

(a) PRMs based on Llama 3.1 8B  (b) PRMs based on Qwen 2.5 7B

Figure 2: Best-of-5 results across 5 categories of 12 reasoning benchmarks. FOVER generates synthetic training data (FOVER-80K) from formal reasoning verification. PRMs trained on FOVER-80K (Ours) generalize to diverse reasoning tasks and consistently outperform baseline PRMs based on the original LLM (Original). Full results are in Table 3.

## 2 RELATED WORK

**Process Reward Models.** PRMs are models that provide feedback for each step in reasoning traces. Compared to Outcome Reward Models (ORMs), which generate a score for the entire solution, PRMs provide fine-grained feedback and often achieve better performance in downstream applications (Uesato et al., 2022; Lightman et al., 2024). PRMs are typically created by fine-tuning LLMs on the task of detecting errors in LLM reasoning at the step level, and a key challenge lies in obtaining step-level error labels for training data. Human annotation (Uesato et al., 2022; Lightman et al., 2024) is a primary approach, but it is costly in this step-level task. Recent studies explore synthetic annotations, such as perturbations (Yang et al., 2022; Ma et al., 2023; Paul et al., 2024) and Monte Carlo roll-outs (Wang et al., 2024a; Luo et al., 2024). However, they can produce unnatural or inaccurate error labels, resulting in noisy training data. In contrast, FOVER leverages formal verification tools to automatically and accurately annotate step-level error labels for PRM training data.

**Training LLMs with symbolic tasks.** Recent work shows that training data created with symbolic reasoning tasks enhances the general reasoning abilities of LLMs even on out-of-distribution tasks (Morishita et al., 2024; Xie et al., 2025). These studies motivate us to hypothesize that LLM-based PRMs can acquire generalizable verification capabilities using training data created with FOVER, which includes error labels on symbolic tasks.

**Formal logic.** Formal logic tasks, such as logical entailment, have been used for evaluating (Clark et al., 2021; Tafjord et al., 2021) or improving (Morishita et al., 2024) reasoning capabilities of LLMs. However, much of prior work uses these tasks without fully leveraging their verifiability, despite the availability of automatic solvers such as Z3 (de Moura & Bjørner, 2008), Vampire (Kovács & Voronkov, 2013), and E (Schulz, 2002), which can verify logical correctness. Our work is the first to use these tools for step-level verification of reasoning traces on formal logic tasks.

**Formal theorem proving.** Using LLMs as a tool for automatic theorem proving is a popular research topic (Polu & Sutskever, 2020; Yang et al., 2023; Xin et al., 2024). Recent work also uses formal theorem proving to improve the math reasoning of LLMs during inference (Zhou et al., 2024). These studies use proof assistants such as Isabelle/HOL (Nipkow et al., 2002), Coq (The Coq Development Team, 2024), and Lean (Moura & Ullrich, 2021) to provide solution-level feedback. Our work is the first to use these tools to annotate step-level error labels on formal proofs.

## 3 FOVER

We propose FOVER, a method to create PRM training data using step-level error labels automatically annotated by formal verification tools, such as Z3 and Isabelle. FOVER creates PRM training data that includes accurate step-level error labels on LLM responses to symbolic logic tasks, without relying on human annotation. For example, we can create PRM training data with first-order logical reasoning tasks using Z3 and with formal theorem proving tasks using Isabelle.

### 3.1 STEP-LEVEL FORMAL VERIFICATION

**Background: Formal verification in formal reasoning tasks.** Formal reasoning tasks (or symbolic reasoning tasks), including formal logic and formal theorem proving, are problems defined and

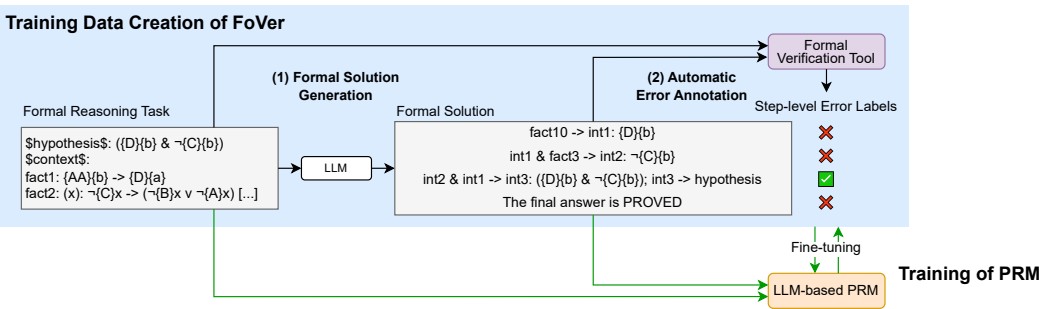

Figure 3: Top: FOVER creates PRM training data using formal verification tools. Bottom: Using the training data, we fine-tune PRMs on the task of step-level verification.

solved using formal syntax and rules (Nawaz et al., 2019; Clark et al., 2021). Formal verification tools, such as automated solvers and theorem provers, are used to formally verify solutions to formal reasoning tasks (Zhou et al., 2024; Yang et al., 2024). Formal verification tools differ in the methods they use and in the groups of tasks to which they apply. For example, Z3 (de Moura & Bjørner, 2008) is an SMT solver, which extends a SAT solver with background theories, applicable to decidable subsets of first-order logic. Isabelle/HOL (Nipkow et al., 2002) is an interactive theorem prover applicable to higher-order logic.

To illustrate the process of formal verification, we present an example of a first-order reasoning task (logical entailment) that can be verified using a SAT solver:

$$\text{Context: } A \rightarrow B, \ B \rightarrow C, \ A \qquad \text{Hypothesis: } C$$

Assume that the predicted answer is 'entailment," meaning the hypothesis is logically entailed by the context. We want to verify whether this prediction is correct, which is equivalent to showing that the following implication is valid:

$$\big((A \rightarrow B) \wedge (B \rightarrow C) \wedge A\big) \ \rightarrow \ C.$$

We can check this with a SAT solver using resolution: combine the premises with the negation of the hypothesis and test its satisfiability in Conjunctive Normal Form (CNF):

$$(\neg A \vee B) \wedge (\neg B \vee C) \wedge A \wedge \neg C.$$

If the solver reports this formula unsatisfiable, then the entailment is confirmed. SAT solvers such as Z3 automate this process and thus can formally verify the correctness of the prediction.

**Step-level formal verification.** In this paper, we propose using formal verification tools to verify solutions at the step level. As illustrated above, these tools have primarily been applied to solution-level verification. Our key idea is that, for certain tasks, they can also be employed to verify the correctness of individual reasoning steps. For instance, for the above problem, suppose the model produces the following step-by-step solution:

$$\text{Step 1:} \quad \big((B \rightarrow C) \wedge A\big) \ \rightarrow \ B. \qquad \text{Step 2:} \quad \big(B \wedge (B \rightarrow C)\big) \ \rightarrow \ C.$$

It can be verified at the step level: we adopt a simple strategy that evaluates the logical correctness of each step independently, assuming that all preceding intermediate results are correct. Once we check that each step only uses the provided context and preceding results, we can verify it by testing the satisfiability of:

$$\text{Step 1:} \quad (\neg B \vee C) \wedge A \wedge \neg B. \qquad \text{Step 2:} \quad B \wedge (\neg B \vee C) \wedge \neg C.$$

The SAT solver will show that the first formula is satisfiable, indicating Step 1 is incorrect, and the second is unsatisfiable, indicating Step 2 is correct. This procedure provides step-level verification for the solution to the logical entailment task.

In this work, we implement step-level verification for formal logic (logical entailment) using Z3 and theorem-proving tasks using Isabelle to create PRM training data.

Table 2: Statistics of the FOVER-80K dataset.

| Tasks | Source Datasets | Formal Verification Tools | Step-by-step Solutions | Step-level Error Labels | |
|---|---|---|---|---|---|
| | | | | # of Steps | % Error |
| Formal Logic | FLDx2 | Z3 | Llama 3.1 8B | 20,000 | 50% |
| | | | Qwen 2.5 7B | 20,000 | 50% |
| Formal Theorem Proving | GSM8K, MetaMathQA, Big-Math | Isabelle | Llama 3.1 8B | 20,000 | 50% |
| | | | Qwen 2.5 7B | 20,000 | 50% |

**Formal Logic**

Formal Statement
```
$hypothesis$: ({D}{b} & ¬{C}{b})
$context$:
fact1:  {AA}{b} -> {D}{a}
fact2:  (x): ¬{C}x -> (¬{B}x v ¬{A}x)  [...]
fact10: {A}{b} -> ({D}{b} & ¬{C}{b})  [...]
```

Formal Solution

| | |
|---|---|
| fact10 -> int1: {D}{b} | ✗ |
| int1 & fact3 -> int2: ¬{C}{b} | ✗ |
| int2 & int1 -> int3: ({D}{b} & ¬{C}{b}); int3 -> hypothesis | ✅ |
| The final answer is PROVED | ✗ |

**Formal Theorem Proving**

Formal Statement
```
assumes "(total_pencils::nat) = 1500"
    and "(cost_per_pencil::real) = 0.10"
    and "(sell_price_per_pencil::real) = 0.25"
    and "(profit::real) = 100.00 [...]"
shows "pencils_to_sell = 1000"
```

Formal Solution

| | |
|---|---|
| have "total_pencils * cost_per_pencil = 150" | ✅ |
| then have "profit + (total_pencils * cost_per_pencil) = 250" | ✅ |
| then have "(profit + (total_pencils * cost_per_pencil)) / [...]" | ✅ |
| then have "(profit + (total_pencils * cost_per_pencil)) / sell_price_per_pencil = 1000" | ✅ |
| have "total_pencils * cost_per_pencil / sell_price_per_pencil = 600" | ✗ |
| then have "pencils_to_sell = 1000" | ✅ |

Figure 4: Examples from FOVER-80K.

## 3.2 FOVER FRAMEWORK

As shown in Figure 3, FOVER synthesizes PRM training data in two stages. In the first stage, given a problem $p \in \mathcal{P}$ from a symbolic task that is compatible with formal verification, an LLM generates a step-by-step formal solution $s = [s_1, s_2, \ldots, s_k] \in \mathcal{S}$ consisting of $k$ steps:

$$[s_1, s_2, \ldots, s_k] = \text{LLM}(p).$$

The solution $s$ may contain logical errors but is required to follow the format compatible with a formal verification tool. We can make LLMs to generate formal solutions either by specifying the required formats in prompts or by using models trained for specific formal verification tools (Yang et al., 2023; Xin et al., 2024). In this work, we provide a few-shot demonstration to guide LLMs. LLMs can generate solutions in an invalid format, so we generate multiple solutions until we obtain one with a valid format, which can be verified using the formal verification tool.

In the second stage, the formal verification tool assigns step-level error labels $[y_{s_1}, y_{s_2}, \ldots, y_{s_k}] \in \{0, 1\}^K$, without the need for human annotation:

$$[y_{s_1}, y_{s_2}, \ldots, y_{s_k}] = \text{FormalVerification}(p, [s_1, s_2, \ldots, s_k]),$$

where FormalVerification(, ) accurately outputs label $y_{s_k} = 1$ if the solution step $s_k$ is correct, $y_{s_k} = 0$ otherwise.

Using training data created with FOVER, we train PRMs ($\mathcal{P} \times \mathcal{S} \to [0, 1]^K$) on a step-level binary classification task, which is the standard approach for training PRMs (Wang et al., 2024a; Zhang et al., 2025). The cross-entropy training objective for PRMs is given by:

$$\mathcal{L}_{\text{PRM}} = \sum_{i=1}^{K} \left( y_{s_i} \log r_{s_i} + (1 - y_{s_i}) \log(1 - r_{s_i}) \right),$$

where $r_{s_i}$ is the step-level score for step $s_i$ predicted by the PRM. In our implementation, we fine-tune LLMs to serve as PRMs, training them in a text generation task to output the token "correct" or "wrong" for each step.

## 3.3 FOVER-80K DATASET

To demonstrate the practical effectiveness of FOVER, we synthesize PRM training data that includes step-level binary error labels on the formal logic and formal theorem proving tasks, which we refer to

as FOVER-80K. As shown in Table 2, FOVER-80K includes 80K steps in reasoning traces generated by Llama 3.1 8B (Llama Team, 2024) and Qwen 2.5 7B (Qwen Team, 2024) with error labels annotated by formal verification tools, Z3 and Isabelle. Figure 4 shows examples from the dataset.

Specifically, the FOVER-80K dataset is created using FLDx2 (Morishita et al., 2023; 2024) and GSM8K-level mathematical reasoning tasks. We select these tasks based on diversity, simplicity, and generalization considerations. For the formal logic task, we selected FLDx2 among logic datasets because it offers the greatest diversity among those expressible via deduction rules (Morishita et al., 2024). For mathematical reasoning, we selected GSM8K-level datasets to simplify the verification pipeline. The choices of these simple tasks also allow us to evaluate the easy-to-difficult generalization of our approach (§4.5).

**Formal logic.** We use the logical entailment task in FLDx2 (Morishita et al., 2023; 2024), a dataset for multi-step first-order logic deduction, in which the goal is to determine whether a hypothesis is entailed by a given set of premises. We generate step-by-step formal solutions with the LLMs and annotate step-level error labels using Z3.

**Formal theorem proving.** We use the task of formal theorem proving for verifying solutions for math word problems (Wu et al., 2022; Zhou et al., 2024). Specifically, the conditions of a math word problem are presented as premises, a candidate final answer is formulated as a hypothesis, and the task is to generate a proof for this statement. We generate formal proofs with the LLMs in the task of verifying the solutions to GSM8K-level problems, including GSM8K (Cobbe et al., 2021), GSM8K-based cases in MetaMathQA (Yu et al., 2024), and math word problems in Big-Math (Albalak et al., 2025). We then annotate step-level error labels using Isabelle.

## 4 EXPERIMENTS

We evaluate LLM-based PRMs trained on FOVER-80K on broad reasoning benchmarks. We compare our PRMs with various PRMs in Best-of-K on 12 reasoning benchmarks (§4.2) and step-level verification on ProcessBench (Zheng et al., 2025) (§4.3). The results are summarized as follows:

- LLM-based PRMs trained on FOVER-80K significantly improve verification capabilities on a broad range of reasoning tasks compared to baseline PRMs based on the same LLMs without additional training, demonstrating effective cross-task generalization of FOVER (§4.2, 4.3).

- LLM-based PRMs trained on FOVER-80K are competitive with or better than state-of-the-art PRMs. The state-of-the-art PRMs, which are trained on math datasets, tend to perform better on math reasoning tasks, but FOVER-PRMs often outperform them on other tasks. This result suggests that FOVER is effective at improving PRMs on out-of-distribution tasks (§4.2, 4.3).

- We evaluate PRMs trained on a combination of FOVER and PRM800K (Lightman et al., 2024), and show that FOVER can effectively complement other datasets (§4.4).

### 4.1 EXPERIMENTAL SETTING

**Datasets.** We use two evaluation tasks. First, we evaluate PRMs using **Best-of-K**, a popular approach to assess PRMs (Cobbe et al., 2021; Li et al., 2023; Zhang et al., 2024b). In Best-of-K, PRMs select the best solution from multiple candidates generated by LLMs for the same input. We evaluate Best-of-K performance of PRMs on 12 reasoning datasets, including math reasoning: GSM8K (Cobbe et al., 2021), MATH (Hendrycks et al., 2021), AQuA-RAT (Ling et al., 2017), AIME (2016-2024) (Art of Problem Solving, 2025), logic reasoning: FOLIO (Han et al., 2024), LogicNLI (Tian et al., 2021), NLI: ANLI (Nie et al., 2020), HANS (McCoy et al., 2019), MMLU: MMLU-Pro-NoMath (Paech, 2024), BIG-Bench Hard: temporal sequences, tracking shuffled objects (three objects), word sorting (Suzgun et al., 2023) (§4.2). For large datasets, we use 250 randomly sampled examples from each dataset for evaluation. Second, we evaluate **step-level verification** performance on ProcessBench (Zheng et al., 2025) (§4.3).

**PRMs.** To demonstrate the applicability of FOVER, we use two LLMs, Llama 3.1 8B (Llama Team, 2024) and Qwen 2.5 7B (Qwen Team, 2024), as base LLMs for PRMs. We fine-tunes these LLMs on FOVER-80K training data of its own responses, which we refer to as **FOVER-PRMs**. **Baseline PRMs:** We compare with baselines that use the base LLMs as PRMs. **State-of-the-art PRMs:** We

Table 3: Best-of-K (K=5) performance of PRMs. FOVER-PRMs (the bottom row) significantly outperform the baseline PRMs in the first row. Compared to the state-of-the-art PRMs in the following rows, FOVER-PRMs are frequently better in tasks other than math reasoning, on which the state-of-the-art PRMs are trained on. $\mu$: Trained on math reasoning tasks. $\dagger$: Trained on human annotated labels. $\ddagger$: Trained on labels annotated by stronger models. $*$: Statistically significant improvement over the baseline in the first row ($p < 0.05$, paired bootstrap (Koehn, 2004)). Best values in each column are shown in bold.

(a) PRMs based on Llama 3.1 8B select the best solution from $K = 5$ solutions generated by Llama 3.1 8B.

| PRMs | Math | | | | Logic | | NLI | | MMLU | BBH | | | Average |
|---|---|---|---|---|---|---|---|---|---|---|---|---|---|
| | GSM8K | MATH | AQuA | AIME | FOLIO | LogicNLI | ANLI | HANS | Pro-NoMath | Temporal | Tracking | Sorting | |
| Llama 3.1 8B | 86.8 | 42.4 | 65.0 | 3.2 | 57.6 | 38.8 | 27.2 | 73.6 | 56.4 | 90.0 | 90.0 | **40.0** | 55.9 |
| RLHFlow-Llama3.1-8B-M$^{\mu\ddagger}$ | **92.0**$^*$ | 45.2 | 64.6 | 2.8 | 59.1 | 44.0 | **29.2** | 79.2$^*$ | 54.0 | 92.0 | 91.2 | 38.0 | 57.7$^*$ |
| RLHFlow-Llama3.1-8B-D$^{\mu\ddagger}$ | 91.6$^*$ | **46.8**$^*$ | 67.7 | **4.4** | 60.6 | 39.6 | **29.2** | 76.0 | 57.2 | **98.8**$^*$ | 92.0 | 38.4 | 58.5$^*$ |
| FOVER-Llama3.1-8B-PRM (ours) | 86.4 | 43.2 | 65.7 | 4.0 | **64.0**$^*$ | **44.8**$^*$ | 28.8 | **82.8**$^*$ | 57.2 | 97.6$^*$ | **93.2**$^*$ | 38.4 | **58.8**$^*$ |

(b) PRMs based on Qwen 2.5 7B select the best solution from $K = 5$ solutions generated by Qwen 2.5 7B.

| PRMs | Math | | | | Logic | | NLI | | MMLU | BBH | | | Average |
|---|---|---|---|---|---|---|---|---|---|---|---|---|---|
| | GSM8K | MATH | AQuA | AIME | FOLIO | LogicNLI | ANLI | HANS | Pro-NoMath | Temporal | Tracking | Sorting | |
| Qwen 2.5 7B | 90.4 | 64.4 | 79.5 | 12.0 | 62.1 | 46.8 | 32.0 | 84.8 | 58.8 | 90.0 | 89.2 | 28.8 | 61.6 |
| Qwen2.5-7B-Skywork-PRM$^{\mu}$ | **96.0**$^*$ | 68.8$^*$ | 78.0 | 13.6 | 62.6 | 44.8 | 30.0 | 82.0 | 60.0 | 82.4 | 90.8 | 30.8 | 61.6 |
| Qwen2.5-Math-7B-PRM800K$^{\mu\dagger}$ | 94.8$^*$ | 68.8$^*$ | 81.1 | 13.2 | **66.5**$^*$ | 47.6 | **34.0** | 83.6 | 58.8 | 81.6 | 90.4 | 28.4 | 62.4 |
| Qwen2.5-Math-PRM-7B$^{\mu\ddagger}$ | 94.8$^*$ | **69.2**$^*$ | **82.7** | **15.2**$^*$ | 65.5 | 48.8 | 28.0 | 84.8 | 58.4 | 84.4 | **92.0**$^*$ | 29.6 | 62.8$^*$ |
| FOVER-Qwen2.5-7B-PRM (ours) | 92.8$^*$ | 66.4 | 82.3 | 13.2 | 65.5 | 48.8 | 32.4 | **85.2** | **60.8** | **91.6** | 89.6 | **31.2** | **63.3**$^*$ |

Table 4: Step-level binary classification performance of PRMs on ProcessBench (AUROC). The first two rows represent PRMs based on LLMs without additional training. The next set of rows contains PRMs trained on math reasoning tasks. The last row shows our FOVER-PRMs. FOVER-PRMs exhibit easy-to-difficult generalization, improving verification performance on tasks that are more complex than those seen during training. $*$: Statistically significant improvement over the baseline PRMs in the first row ($p < 0.05$, paired bootstrap).

<table>
<tr><td colspan="6" align="center">(a) Llama 3.1 as PRMs</td><td colspan="6" align="center">(b) Qwen 2.5 as PRMs</td></tr>
<tr><td>PRMs</td><td>GSM8K</td><td>MATH</td><td>Olympiad</td><td>Omni</td><td>Ave.</td><td>PRMs</td><td>GSM8K</td><td>MATH</td><td>Olympiad</td><td>Omni</td><td>Ave.</td></tr>
<tr><td>Llama 3.1 8B</td><td>70.9</td><td>68.8</td><td>67.3</td><td>59.0</td><td>66.5</td><td>Qwen 2.5 7B</td><td>77.9</td><td>76.0</td><td>74.9</td><td>73.6</td><td>75.6</td></tr>
<tr><td>Llama 3.1 70B</td><td>92.0</td><td>82.2</td><td>83.4</td><td>80.5</td><td>84.5</td><td>Qwen 2.5 72B</td><td>87.5</td><td>83.5</td><td>81.8</td><td>80.9</td><td>83.4</td></tr>
<tr><td>RLHFlow-Llama3.1-8B-M$^{\mu\ddagger}$</td><td>91.2$^*$</td><td>75.2$^*$</td><td>68.2</td><td>60.7</td><td>73.8$^*$</td><td>Qwen2.5-7B-Skywork-PRM$^{\mu}$</td><td>92.3$^*$</td><td>82.6$^*$</td><td>72.6</td><td>64.8</td><td>78.1</td></tr>
<tr><td>RLHFlow-Llama3.1-8B-D$^{\mu\ddagger}$</td><td>88.5$^*$</td><td>77.2$^*$</td><td>73.6$^*$</td><td>63.1$^*$</td><td>75.6$^*$</td><td>Qwen2.5-Math-7B-PRM800K$^{\mu\dagger}$</td><td>94.3$^*$</td><td>91.7$^*$</td><td>91.5$^*$</td><td>90.3$^*$</td><td>92.0$^*$</td></tr>
<tr><td>FOVER-Llama3.1-8B-PRM (ours)</td><td>80.0$^*$</td><td>74.1$^*$</td><td>74.8$^*$</td><td>74.7$^*$</td><td>75.9$^*$</td><td>Qwen2.5-Math-PRM-7B$^{\mu\ddagger}$</td><td>97.7$^*$</td><td>95.3$^*$</td><td>94.8$^*$</td><td>93.2$^*$</td><td>95.3$^*$</td></tr>
<tr><td></td><td></td><td></td><td></td><td></td><td></td><td>FOVER-Qwen2.5-7B-PRM (ours)</td><td>90.8$^*$</td><td>89.1$^*$</td><td>84.6$^*$</td><td>86.0$^*$</td><td>87.6$^*$</td></tr>
</table>

also compare with five state-of-the-art PRMs that are built on the same LLMs. Among PRMs based on Llama 3.1 8B, we evaluate RLHFlow-Llama3.1-8B trained on the DeepSeek or Mistral data (Xiong et al., 2024), which include error labels on solutions generated by stronger models on GSM8K and MATH acquired via Monte Carlo roll-outs (Wang et al., 2024a). Among PRMs based on Qwen 2.5 7B, we evaluate Qwen2.5-Math-7B-PRM800K (Zheng et al., 2025), which is trained on human-annotated labels on MATH, and Qwen2.5-Math-PRM-7B (Zhang et al., 2025), which is trained on labels synthesized using Monte Carlo roll-outs and verification by a stronger model. We also evaluate Qwen2.5-7B-Skywork-PRM (He et al., 2024), which is trained on math and coding.

**Implementation Details.** We create inputs to LLM-based PRMs by preprocessing step-by-step solutions into a conversation format where each input contains a single step, and the expected output is a single token: "correct" or "incorrect". To obtain step-level scores, we extract logits for the two words and apply the softmax function to compute the prediction probability for "correct". This is a popular approach to use LLMs as PRMs (Xiong et al., 2024). As the baseline PRMs are not fine-tuned, we provide zero-shot instructions about this format. Refer to Appendix H for details.

## 4.2 RESULTS OF BEST-OF-K ON REASONING

For each input, we first use a temperature of 0.5 and few-shot demonstrations to generate $K = 5$ solutions from Llama 3.1 8B and Qwen 2.5 7B. The few-shot demonstrations enable models to generate solutions in a step-by-step format. Following existing work (Lightman et al., 2024; Wang et al., 2024a), we then use PRMs to score each step in a solution, and the solution score is the minimum score across all steps. The solution with the highest solution score is selected.

Table 3 compares the Best-of-K performance of PRMs. Remarkably, our FOVER-PRMs trained on FOVER-80K—consisting of formal logic and theorem proving tasks—significantly improve performance on broad out-of-distribution reasoning tasks compared to the baseline PRMs. This result demonstrates that our training data created with FOVER, which includes error labels annotated by formal verification tools on symbolic tasks, exhibits cross-task generalization and enhances PRMs across a broad range of informal reasoning tasks written in natural language.

We also observe that FOVER-PRMs achieve performance competitive with that of state-of-the-art PRMs. The state-of-the-art PRMs, which are trained to detect mistakes in mathematical reasoning, tend to perform better on math reasoning, as expected. However, FOVER-PRMs often outperform them on other reasoning tasks that are out-of-distribution for both FOVER and the state-of-the-art PRMs. This result suggests that FOVER is advantageous for generalization and is effective in improving PRMs on out-of-distribution reasoning tasks.

## 4.3 RESULTS OF STEP-LEVEL VERIFICATION ON PROCESSBENCH

We evaluate PRMs on the step-level binary classification task in ProcessBench (Zheng et al., 2025), which includes human-annotated step-level error labels (correct vs. incorrect) for responses from multiple LLMs on four math reasoning tasks (GSM8K, MATH, Olympiad-Bench, and Omni-Math). ProcessBench includes labels only for the earliest error in each response, so we evaluate PRMs on steps up to the first error in each solution.

Table 4 shows the step-level verification performance of PRMs, measured in AUROC. The result shows that training on FOVER-80K, based on GSM8K-level problems, significantly improves step-level verification performance in more complex math reasoning tasks (MATH, Olympiad-Bench, and Omni-Math) when compared to the baseline PRMs. Furthermore, we observe that FOVER-PRM based on Llama 3.1 8B is competitive to PRMs that are trained on GSM8K and MATH, and FOVER-PRM based on Qwen 2.5 7B is better than PRMs based on Qwen 2.5 72B. These results demonstrate that training on FOVER-80K exhibits easy-to-difficult generalization, improving PRMs on reasoning tasks that are more complex than those seen during training.

## 4.4 COMBINING FOVER WITH EXISTING DATASETS

FOVER offers a unique approach for creating PRM training data using tasks different from those used in existing methods like Monte Carlo roll-outs (Wang et al., 2024a), which are mainly designed for math reasoning tasks. This enables our training data to complement other datasets, increasing the diversity of PRM training data. To demonstrate its practical effectiveness when combined with other datasets, we evaluate PRMs trained on a combination of FOVER-80K and PRM800K (Lightman et al.,

Table 5: Best-of-K performance of PRMs trained on FOVER-80K and PRM800K.

| Training Data | Llama 3.1 8B | | | Qwen 2.5 7B | | |
|---|---|---|---|---|---|---|
| | GSM8K | MATH | HANS | GSM8K | MATH | HANS |
| FOVER-80K | 86.4 | 43.2 | 82.8 | 92.8 | 66.4 | 85.2 |
| PRM800K | 89.2 | 41.2 | 80.0 | 92.8 | 64.8 | 85.2 |
| FOVER-80K + PRM800K | **90.0** | **44.8** | **83.2** | **93.6** | **66.8** | **85.6** |

2024). As shown in Table 5, PRMs trained on FOVER-80K and PRM800K are often competitive with or better than PRMs trained on either dataset alone, especially on math reasoning tasks. This result indicates that FOVER can be effectively used in combination with existing datasets.

## 4.5 GENERALIZATION IN PRMS TRAINED ON FOVER

FOVER creates PRM training data using symbolic tasks, whose properties differ from those of popular tasks to which PRMs are applied. However, our results demonstrate that PRMs trained with

Table 6: Case study of the generalization of FOVER-PRMs to out-of-distribution tasks.

(a) Verification of an LLM response to the temporal sequence task in BBH by PRMs based on Qwen 2.5 7B. FOVER-PRM successfully detects the wrong step (Step 7), while the baseline PRM fails.

> **Problem:** Today, Thomas went to the football field. Between what times could they have gone? Thomas woke up at 8am. [...] Lisa saw Thomas walking towards the Statue of Liberty from 12pm to 2pm. [...] (A) 12pm to 2pm [...] (D) 5pm to 6pm
> **Ground truth answer:** (D) 5pm to 6pm

| Step | Solution steps generated by Qwen 2.5 7B | Baseline | FOVER |
|---|---|---|---|
| Step 3 | Lisa saw Thomas walking towards the Statue of Liberty from 12pm to 2pm, so Thomas did not go to the football field from 12pm to 2pm. | 1.00 | 0.98 |
| Step 7 | Im summary, the only time when Thomas could have gone to the football field was 12pm to 2pm. | **0.96** | **0.38** |
| Step 9 | So the answer is (A). | 0.98 | 1.00 |

(b) Verification of an LLM response to MMLU-Pro by PRMs based on Llama 3.1 8B. FOVER-PRM successfully detects the wrong step (Step 1), while the baseline PRM fails.

> **Problem:** Nagel clarifies that absolutism is the view that we should (A) [...] (B) never prevent murder [...] (I) avoid murder at all costs
> **Ground truth answer:** (I) avoid murder at all costs

| Step | Solution steps generated by Llama 3.1 8B | Baseline | FOVER |
|---|---|---|---|
| Step 0 | [...] In the context of morality, absolutism is the view that a particular action, such as murder, is always right or always wrong, regardless of the circumstances. | 0.84 | 0.91 |
| Step 1 | Option (B) "never prevent murder" is the correct interpretation of absolutism in this context, as it implies that murder is always wrong and should never be prevented, which is a characteristic of an absolutist view. | **0.99** | **0.47** |
| Step 2 | The answer is (B). | 0.98 | 0.99 |

FOVER generalize effectively, showing its practical usefulness. First, FOVER exhibits **symbol-to-text generalization**. FOVER-80K includes symbolic solutions that are compatible with Z3 and Isabelle, but training on this dataset improves PRMs on informal tasks whose solutions are written in natural language. Furthermore, our results show **easy-to-difficult generalization**. Our dataset, created using GSM8K-level problems, improves PRMs on more complex math reasoning tasks, including MATH, Olympiad-Bench, and Omni-Math (§4.3). Finally, FOVER exhibits **cross-task generalization** and improves PRMs on tasks that are largely different from those in training data (§4.2). Surprisingly, FOVER improves verification performance of PRMs on MMLU-Pro and BBH, whose properties differ substantially from those of our training tasks.

Table 6a shows an example from the temporal sequence task in BBH. The task is to find a free time slot that does not overlap with any events in a given list of past time intervals, which is distinct from logical reasoning and theorem proving tasks in FOVER-80K. This example shows that FOVER-PRM properly understands the task and solutions described in natural language and correctly detects a logical error in Step 7. Table 6b shows an example from MMLU-Pro. In this example, FOVER-PRM correctly understands the concept of "absolutism" in Step 0 and detects a logical mistake in Step 1. Although FOVER-80K does not include knowledge-intensive tasks, it makes PRMs generalize to tasks that require reasoning over domain knowledge. These examples demonstrate effective text-to-symbol and cross-task generalization in PRM training with FOVER.

## 5 CONCLUSION

We introduce FOVER, a method to create PRM training data using step-level error labels annotated by formal verification tools. In contrast to existing methods that rely on human annotation or inaccurate synthetic labels, FOVER is the first method to automatically generate accurate PRM training data. Using FOVER, we create PRM training data with accurate error labels on formal logic and theorem proving tasks. While FOVER generates PRM training data using symbolic tasks that can be formally verified, PRMs trained on our dataset improve the verification capabilities across broad out-of-distribution reasoning tasks, showing its practical usefulness.

ETHICS STATEMENT

This research does not involve human subjects or private data. This paper proposes a method for improving PRMs, which detect mistakes made by LLMs. We do not expect any harmful impact.

REPRODUCIBILITY STATEMENT

We provide code and datasets in the supplementary material and provide detailed settings in the appendix. We will release these materials and models evaluated in this paper to the public.

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

## TABLE OF CONTENTS OF APPENDIX

## A   THE USE OF LARGE LANGUAGE MODELS

In paper writing, we used LLMs to polish writing and for retrieval and discovery (e.g., finding related work). Specifically, we used ChatGPT-4o and ChatGPT-5 via OpenAI's web interface.

## B   LIMITATIONS

**Model size.** Following PRMs introduced in prior work (Zheng et al., 2025; Zhang et al., 2025; He et al., 2024), our experiments focus on PRMs based on LLMs with less than 8B parameters.

**Tasks in FOVER-80K.** Our training data in this paper targets improving PRMs at the verification of responses from 8B-class LLMs and includes tasks that may not be difficult for larger LLMs. When applying our approach to improve PRMs based on stronger models, we need to use more difficult tasks to make the larger LLMs introduce meaningful mistakes. This is not a limitation of FOVER, and we can create training data using more complex tasks with FOVER. Appendix D discusses how we can apply FOVER to create PRM training data based on more complex tasks.

## C   ADDITIONAL RELATED WORK

**PRMs for non-mathematical tasks.** Similar to our work, recent work has started working on PRMs for non-mathematical tasks. Zeng et al. (2025) train PRMs on their dataset created using MMLU-Pro (Wang et al., 2024b) by annotating error labels using strong LLMs and show performance improvement in diverse domains like law and philosophy. Yin et al. (2025) propose a method to adaptively select evaluation criteria from a reward tree that includes evaluation criteria extracted from training data. Our work, which demonstrates that training data generated with formal verification can effectively leverage the inherent generalization capability of LLM-based PRMs, is orthogonal to these studies.

**Outcome-based training for PRMs.** Provided the difficulty of collecting step-level error labels, recent work proposes methods to train PRMs without using step-level labels. Yuan et al. (2025) proposes implicit PRMs that can be trained only using final answers. They theoretically show that ORMs trained with the reward that is parameterized by the log-likelihood ratio of two causal language models (e.g., DPO (Rafailov et al., 2023)) implicitly learns a Q function and can be used as PRMs. In addition, recent work (Feng et al., 2025; Kim et al., 2025) reports that existing Large Reasoning Models like DeepSeek-R1 (DeepSeek-AI, 2025) have strong process-level rewarding capabilities on mathematical reasoning tasks, while they are not explicitly trained for process-level rewarding. However, these studies rely on outcome-based reward and can only be applied to tasks like mathematical reasoning. FOVER, which enables training data creation using tasks other than mathematical reasoning, is orthogonal to these recent studies and can also be used in combination with them to increase the diversity of training data for PRMs.

**Applications of PRMs.** PRMs can be used to supervise LLM reasoning during training and inference. For **training**, PRMs can generate reward signals, particularly in reinforcement learning settings (Pan et al., 2023; Zhang et al., 2024a). They can be applied either to re-rank candidate responses from the policy or to provide direct reward (Uesato et al., 2022). For **inference**, PRMs can guide response selection and refinement through Best-of-K (Li et al., 2023), self-correction (Saunders et al., 2022; Madaan et al., 2023), and step-level search (Ma et al., 2023; Snell et al., 2025).

**Cross-task generalization in LLMs.** Our work is the first to conduct an in-depth analysis of cross-task generalization in LLM-based PRMs. However, cross-task generalization in LLMs has been widely studied in general tasks. Early studies of LLMs, such as T5 (Raffel et al., 2020) and FLAN (Wei et al., 2022), already observe their generalization to unseen tasks. Easy-to-hard generalization (Burns et al., 2023; Hase et al., 2024; Sun et al., 2024) is a challenging type of generalization, where LLMs trained on simpler tasks show improved performance on more complex ones. Directly related to our work, recent work (Morishita et al., 2024; Xie et al., 2025) shows that synthetic and symbolic training data can improve general reasoning capabilities of LLMs. These observations motivate our hypothesis that step-level errors annotated by formal verification tools on symbolic logical tasks can improve LLM-based PRMs on diverse out-of-distribution reasoning tasks.

# D  FUTURE WORK: EXTENDING FOVER TO OTHER TASKS AND TOOLS

In this paper, as the first work towards this direction, we evaluate FOVER by creating PRM training data using relatively simple tasks with a minimal pipeline to keep the evaluation focused and clear. Our experiments show that training data created using these relatively simple tasks by FOVER effectively improve PRMs on a broad range of reasoning tasks, demonstrating the effectiveness and generalization ability of FOVER.

However, FOVER is not limited to the tasks and tools we used in this paper and can be extended to create training data using more complex tasks. When applying FOVER to create PRM training data using more complex tasks, there are two potential challenges: generating formal solutions from LLMs in a valid format compatible with formal verification tools and verifying step-level correctness using the tools.

First, formal solutions should be in a syntactically valid format compatible with the formal verification tools we use for verification. Following complex syntactical rules and formats of some formal verification tools, like theorem provers, can be challenging for LLMs. Recent LLMs are increasingly capable of generating formal solutions in valid formats, showing strong performance in first-order logic (Olausson et al., 2023) and a growing ability to produce syntactically valid formal proofs (Ren et al., 2025). We expect future models to further improve their capabilities to generate formal solutions and be more suitable for creating PRM training data with FOVER.

Second, we need to make formal verification tools verify step-level correctness. The tools are often designed for solution-level verification, so we often need to write wrapper code to adapt them for step-level verification, as we did in this paper for creating FOVER-80K. When creating PRM training data using more complex tasks, we may need to further modify the verification pipeline to support new operations.

For example, to keep the verification pipeline simple, we did not use problems that involve assumptions in the formal logic task (FLDx2), such as proofs by contradiction, when creating FOVER-80K. However, we can extend our verification pipeline to support such cases as well. Existing verification tools are already capable of performing solution-level verification for proofs by contradiction. Therefore, we can make use of it to provide step-level verification. When handling assumptions in our framework, the type of mistake that cannot be detected through our current step-independent verification alone is illustrated by the following example, because the step-independent verification assumes that preceding intermediate results are correct:

- Premises: fact1: B; fact2: B→C; fact3: C→A;
- Hypothesis: A
- fact1 and fact2 → C; **Assume A; A → Hypothesis;** Therefore, the hypothesis is proved.

In this case, the existing solution-level verification will identify this solution as an error because the assumption is not properly discharged. Thus, by combining step-independent verification with solution-level verification, we can identify and label the final step as erroneous.

We also explain an approach to extend our verification pipeline to more complex formal theorem proving tasks in Appendix G.2.3.

# E  MODEL ACCESS AND SOFTWARE LIBRARIES

This section provides details of LLMs and libraries used in our experiments.

## E.1  MODELS

We use models that are provided at Hugging Face Hub.

**Base models.** We use `meta-llama/Llama-3.1-8B-Instruct` and `Qwen/Qwen2.5-7B-Instruct` as base models for our PRMs. We also use these models to generate initial responses used in creating FOVER-80K, and also for generating $K = 5$ responses in the Best-of-K evaluation (§4.2).

**Informal-to-formal conversion in the formal theorem proving task.** As explained in Appendix G, we use `meta-llama/Llama-3.3-70B-Instruct` for converting informal statements and solutions to the Isabelle format (autoformalization).

## E.2  STATE-OF-THE-ART PRMS COMPARED

This section provides details of state-of-the-art PRMs we evaluate in Section 4. Table 7 shows the details of the models we evaluate. We acquire these models at Hugging Face Hub and use vLLM (Kwon et al., 2023) to generate reward scores.[1]

Table 7: State-of-the-art PRMs we evaluate in Section 4.2.

| PRMs | Source | Base Datasets | Error Annotation |
|------|--------|---------------|------------------|
| RLHFlow-Llama3.1-8B-DeepSeek (2024) | RLHFlow/Llama-3.1-8B-PRM-Deepseek-Data | GSM8K, MATH | Math-Shepherd (2024a) |
| RLHFlow-Llama3.1-8B-Mistral (2024) | RLHFlow/Llama-3.1-8B-PRM-Mistral-Data | GSM8K, MATH | Math-Shepherd (2024a) |
| Qwen2.5-Math-7B-PRM800K (2025) | Qwen/Qwen2.5-Math-7B-PRM800K | MATH | Human annotation |
| Qwen2.5-Math-PRM-7B (2025) | Qwen/Qwen2.5-Math-PRM-7B | Private Data | Math-Shepherd (2024a) & LLM-as-a-Judge |
| Qwen2.5-7B-Skywork-PRM (2024) | Skywork/Skywork-o1-Open-PRM-Qwen-2.5-7B | Hidden | Hidden |

## E.3  SOFTWARE LIBRARIES

**Inference Code.** We use vLLM (Kwon et al., 2023) for accelerating LLM inference.

**Training Code.** We use LLaMA-Factory (Zheng et al., 2024) for training.

---

[1] `https://docs.vllm.ai/en/latest/models/supported_models.html#reward-modeling-task-reward`

# F    DETAILS OF FOVER-80K

This section provides details of our FOVER-80K dataset.

## F.1    STATISTICS OF RAW VERIFICATION RESULTS

Table 8 shows the statistics of the raw verification results before balancing the label distribution. To create FOVER-80K, we extracted 40K steps with the balanced label distribution for each model, as shown in Table 2.

Table 8: Statistics of the raw data for FOVER-80K befor balancing the label distribution.

| Responses | Tasks | Train | | | | Validation | | | | Test | | | |
|---|---|---|---|---|---|---|---|---|---|---|---|---|---|
| | | Solution-level | | Step-level | | Solution-level | | Step-level | | Solution-level | | Step-level | |
| | | # | % Error | # | % Error | # | % Error | # | % Error | # | % Error | # | % Error |
| Llama 3.1 8B | Formal Logic | 10000 | 70.0% | 43082 | 43.8% | 360 | 70.0% | 1428 | 44.5% | 360 | 70.0% | 1417 | 45.2% |
| | Formal Proof | 10000 | 70.0% | 69532 | 13.3% | 360 | 70.0% | 2286 | 14.0% | 360 | 70.0% | 2499 | 13.8% |
| Qwen 2.5 7B | Formal Logic | 10000 | 70.0% | 34879 | 41.3% | 360 | 70.0% | 1208 | 39.2% | 360 | 70.0% | 1225 | 41.5% |
| | Formal Proof | 10000 | 70.0% | 69452 | 14.1% | 360 | 70.0% | 2297 | 16.7% | 360 | 70.0% | 2501 | 16.6% |

## F.2    EXAMPLES

We provide examples from FOVER-80K.

**Formal logic.** Here is an example of the data in the formal logic task based on the initial responses from Llama 3.1 8B.

```
{
  'id': 'fldx2-train-058709_Llama-3.1-8B-Instruct',
  'error_labels': [True, False, False, True, False],
  'messages': [
    {
      'role': 'user',
      'content': '** Problem **
       Based on the provided facts ($context$), either prove or disprove the hypothesis or
       ↪  state that it is unknown. The facts and the hypothesis are written in logical
       ↪  formulas as follows: capital letters such as "{A}", "{B}", "{AB}" are predicates,
       ↪  small letters such as "{a}", "{b}", "{ab}" are constants, "&" is logical
       ↪  conjunction, "v" is logical disjunction, "¬" is negation, "->" is implication,
       ↪  "(x)" is "for all x", and "(Ex)" is "for some x".

       $hypothesis$: {AB}{a}

       $context$:
       fact1: (¬{B}{a} & ¬{A}{a}) -> ¬{A}{dk}
       fact2: (¬{AQ}{et} & ¬{DF}{et})
       fact3: ¬{CO}{a} -> (¬{DR}{a} & ¬{CK}{a})
       fact4: ¬{HS}{a}
       fact5: (x): {C}x -> (¬{B}x & ¬{A}x)
       fact6: ¬{JF}{a} -> ¬{AA}{a}
       fact7: (¬{JJ}{du} & ¬{AA}{du})
       fact8: (¬{BB}{a} & ¬{HQ}{a})
       fact9: ¬{AA}{a}
       fact10: ¬{A}{a} -> ¬{AA}{a}
       fact11: ¬{DK}{a}
       fact12: ¬{E}{b} -> ¬(¬{D}{b} & ¬{C}{b})
       fact13: ¬(¬{D}{b} & ¬{C}{b}) -> {C}{a}
       fact14: ¬{A}{ci}
       fact15: ¬{JC}{a}
       fact16: ¬{A}{r} -> ¬{DQ}{r}
       fact17: ¬{A}{a} -> (¬{AA}{a} & ¬{AB}{a})
       fact18: (¬{HP}{a} & ¬{C}{a})
       fact19: ¬{A}{jg}

       ** Task **
       Your task is to evaluate the accuracy of each step in the provided solution to the
       ↪  above question. For each step, respond with "correct" if the reasoning is
       ↪  logically valid and mathematically sound, or if the step is a general statement
       ↪  or transition that does not contain reasoning. Respond with "incorrect" if the
       ↪  step includes any errors or flawed logic.
```

```
        ** Sotluion **
        fact17 & fact10 -> int1: ¬{A}{a} -> ¬{AB}{a}'
    },
    {'role': 'assistant', 'content': 'correct'},
    {'role': 'user', 'content': 'fact10 -> int2: ¬{A}{a}'},
    {'role': 'assistant', 'content': 'incorrect'},
    {'role': 'user', 'content': 'int2 -> int3: ¬{AB}{a}'},
    {'role': 'assistant', 'content': 'incorrect'},
    {'role': 'user', 'content': 'int3 -> ¬hypothesis'},
    {'role': 'assistant', 'content': 'correct'},
    {'role': 'user', 'content': 'The final answer is DISPROVED'},
    {'role': 'assistant', 'content': 'incorrect'}
]
```

**Formal theorem proving.** Here is an example of the data in the formal proof task based on the initial responses from Llama 3.1 8B.

```
{
  'id': 'bigmath_orca_math_138943_Llama-3.1-8B-Instruct',
  'error_labels': [True, False, True, True, True],
  'messages': [
  {
    'role': 'user',
    'content': '** Problem **
      Generate a proof for the following theorem in the Isabelle proof assistant format.

      theorem example:
        assumes "(Stella_Artois_speed::real) = 150"
            and "(total_distance::real) = 500"
            and "(total_time::real) = 1.694915254237288"
            and "(first_car_speed::real) = (total_distance - Stella_Artois_speed *
            ↪  total_time) / total_time"
        shows "first_car_speed = 145.2"

      ** Task **
      Your task is to evaluate the accuracy of each step in the provided solution to the
      ↪  above question. For each step, respond with "correct" if the reasoning is logically
      ↪  valid and mathematically sound, or if the step is a general statement or transition
      ↪  that does not contain reasoning. Respond with "incorrect" if the step includes any
      ↪  errors or flawed logic.

      ** Sotluion **
      have "Stella_Artois_speed * total_time = 150 * total_time"'
  },
  {'role': 'assistant', 'content': 'correct'},
  {
    'role': 'user',
    'content': 'then have "Stella_Artois_speed * total_time = 254.1794318175"
      then have "total_distance - Stella_Artois_speed * total_time = 245.8205681825"'
  },
  {'role': 'assistant', 'content': 'incorrect'},
  {'role': 'user', 'content': 'then have "(total_distance - Stella_Artois_speed * total_time)
  ↪  / total_time = 145.2"'},
  {'role': 'assistant', 'content': 'correct'},
  {'role': 'user', 'content': 'then have "first_car_speed = 145.2"'},
  {'role': 'assistant', 'content': 'correct'},
  {'role': 'user', 'content': 'thus ?thesis'},
  {'role': 'assistant', 'content': 'correct'}
]
```

# G    DATASET CREATION PROCESS OF FOVER-80K

We provide details of the formal logic task in FOVER-80K, which is outlined in Section 3.3.

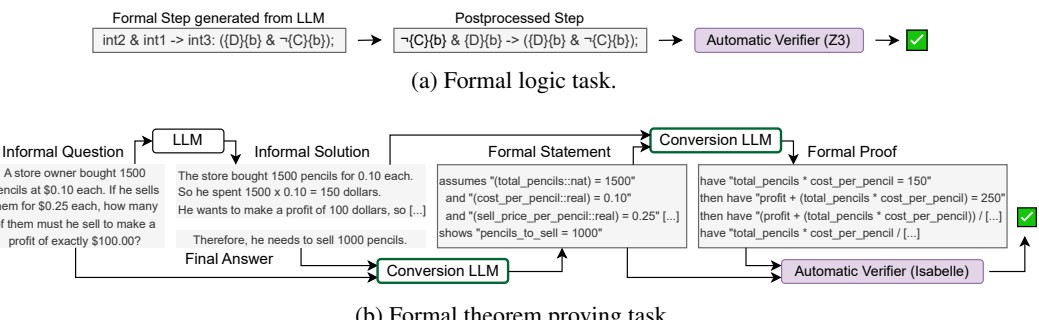

Figure 5: Creation process of FOVER-80K. (1) We first generate LLM reasoning in the format compatible with formal verification tools: Z3 and Isabelle. (2) We use the formal verification tools to automatically annotate step-level error labels, without involving human annotation.

Figure 6: Automatic step-level error annotation for FOVER-80K by formal verification tools.

To create FOVER-80K, we demonstrate two types of training data creation processes of FOVER, as in Figure 5. First, for the formal logic task, we directly generate formal solutions from Llama 3.1 8B and Qwen 2.5 7B, and these models often can generate solutions in a valid format compatible with Z3. Second, in the formal theorem proving task, we find that these small LLMs often generate syntactically invalid proofs because the syntax of Isabelle is complex. Although using stronger LLMs to generate formal proofs directly is a possible option, the resulting training data will include errors whose properties are different from those made by the small LLMs. To show the applicability of FOVER to the situation where we want to train PRMs on mistakes made by small LLMs, we convert informal solutions (i.e., solutions in natural language) generated by the small LLMs to formal proofs using a strong LLM. In both tasks, syntactically invalid solutions are removed using the tools, and all formal solutions in our training data are in a valid format compatible with the verification tools.

FORMAL LOGIC.

**Formal solution generation.** We prompt Llama 3.1 8B and Qwen 2.5 7B to generate step-by-step formal solutions to FLDx2 in a format compatible with Z3. We use a few-shot instruction to guide LLMs to follow the format and filter out syntactically invalid solutions using Z3.

**Automatic error annotation.** Z3 is designed for verification at the solution level, but we use it at the step level by supplying Z3 with the premises and conclusion for the target step to determine logical validity, as in Figure 6a. Refer to Appendix G.1.2 for details.

FORMAL THEOREM PROVING.

**Informal solution generation.** We first generate informal solutions (i.e., solutions in natural language) from Llama 3.1 8B and Qwen 2.5 7B to GSM8K-level problems, including GSM8K (Cobbe et al., 2021), GSM8K-based cases in MetaMathQA (Yu et al., 2024), and math word problems in Big-Math (Albalak et al., 2025). Refer to Appendix G.2.1 for details.

**Informal-to-formal conversion.** We convert the informal solutions into formal proofs using Llama 3.3 70B, as in Figure 6b. We filter out syntactically invalid proofs using Isabelle, so this process generates valid formal proofs compatible with Isabelle. Refer to Appendix G.2.2 for details.

The resulting training data includes pairs of formal proofs and the step-level errors annotated by Isabelle, and we train PRMs on the task of detecting logical mistakes in the formal proofs. Therefore, the informal solutions are not included in training data, and our training data is valid if the error labels are accurate with respect to the formal proofs. While the conversion may occasionally generate formal proofs that are unfaithful to the informal solutions, the accuracy of our training data is not negatively affected because error labels are annotated for the formal proofs using Isabelle.

**Automatic error annotation.** Isabelle is designed for solution-level verification. To obtain step-level error labels, we implement wrapper code for step-level verification. Our code assumes that the other steps are correct when evaluating the target step. Refer to Appendix G.2.3 for details.

### G.1 FORMAL LOGIC

#### G.1.1 BASE DATASET AND INITIAL RESPONSE GENERATION FOR FORMAL LOGIC

First, we need to generate symbolic solutions from LLMs, which may include logical mistakes, but they should be in a valid format compatible with Z3 (de Moura & Bjørner, 2008).

We use FLDx2 (Morishita et al., 2023; 2024) as a base dataset for our formal logic task. We use the symbolic version of the dataset. To simplify the verification pipeline, we removed cases whose reasoning steps include "assump," which is used in cases such as proof by contradiction.

We generate formal solutions from Llama 3.1 8B and Qwen 2.5 7B. The following is an example of a few-shot demonstration for the initial generation. We provide six examples as a demonstration. After generating the formal solutions, we filter out those with an invalid format using Z3.

```
{
  'role': 'user',
  'content':
    'Based on the provided facts ($context$), either prove or disprove the hypothesis or
    ↪   state that it is unknown. The facts and the hypothesis are written in logical
    ↪   formulas as follows: capital letters such as "{A}", "{B}", "{AB}" are predicates,
    ↪   small letters such as "{a}", "{b}", "{ab}" are constants, "&" is logical conjunction,
    ↪   "v" is logical disjunction, "¬" is negation, "->" is implication, "(x)" is "for all
    ↪   x", and "(Ex)" is "for some x".

    Don't generate anything other than the proof and proof_label.
    Don't generate the proof by contradiction. If the hypothesis is disproved, provide a
    ↪   direct counterexample. If the proof_label is PROVED, your proof should end with the
    ↪   'hypothesis'. If the proof_label is DISPROVED, your proof should end with
    ↪   '¬hypothesis'.

    $hypothesis$: ({AB}{c} & ¬{B}{c})

    $context$:
    fact1: (x): ¬{D}x -> ({A}x & {C}x)
    fact2: ¬({B}{a} & ¬{A}{a})
    fact3: ¬{AB}{b} -> ¬{C}{d}
    fact4: ¬{C}{a} -> ¬{A}{d}
    fact5: ¬{A}{a}
    fact6: ({AB}{c} & ¬{B}{c}) -> {C}{d}
    fact7: ¬({F}{b} v {D}{b}) -> ¬{D}{a}
    fact8: {A}{a} -> ({AB}{c} & ¬{B}{c})
    fact9: ({AB}{c} & ¬{AA}{c}) -> {B}{b}
    fact10: ({AB}{c} & ¬{B}{c}) -> {C}{b}'
},
{
  'role': 'assistant',
  'content':
    '$proof$:
    fact6 -> int1: ¬{C}{d} -> ¬({AB}{c} & ¬{B}{c});
    fact3 & int1 -> int2: ¬{AB}{b} -> ¬({AB}{c} & ¬{B}{c});

    $proof_label$: UNKNOWN'
}
```

#### G.1.2 AUTOMATIC STEP-LEVEL VERIFICATION IN FORMAL LOGIC

Second, we automatically annotate step-level error labels, which will be used to train PRMs.

We use Z3 (de Moura & Bjørner, 2008) for step-level verification of the formal logic task. Specifically, we use the checker provided by the FLDx2 paper (Morishita et al., 2024) based on Z3.[2] Z3 is originally designed for solution-level verification, so we need to write code to use it for step-level verification. As explained in Figure 6a, we first postprocess each step in solutions to an independent logical step and check the validity using Z3.

---

[2]https://github.com/hitachi-nlp/FLD-generator/blob/
00d12c4a9132a4fb43cd77f24db03ea7f5b27877/FLD_generator/formula_checkers/
z3_logic_checkers/checkers.py#L179

## G.2 FORMAL THEOREM PROVING

### G.2.1 BASE DATASETS AND INITIAL RESPONSE GENERATION FOR FORMAL PROOF

First, we need to generate symbolic solutions (formal proofs) from LLMs, which may include logical mistakes, but they should be in a valid format compatible with Isabelle (Nipkow et al., 2002). However, we find that it is difficult for Llama 3.1 8B and Qwen 2.5 7B to generate formal theorem proofs in a valid format because Isabelle's syntax is complex. We can generate formal proofs using larger LLMs, but such proofs may include mistakes that have different properties from those made by the small LLMs. To show the applicability of FOVER in the situation where we want to create PRM training data that includes solutions and mistakes generated by the small LLMs, we take an approach to first generate informal solutions from the small models and convert them to formal proofs using a larger model (i.e., autoformalization).

We use informal responses from LLMs to GSM8K-level math word problems: GSM8K (Cobbe et al., 2021), GSM8K-based cases in MetaMathQA (Yu et al., 2024), and math word problems in Big-Math (Albalak et al., 2025). We select these relatively simple math reasoning problems to keep our pipelines for the informal-to-formal conversion and step-level verification simple, but we can also use more complex tasks, as explained in Appendix G.2.3.

The following is an example of a few-shot demonstration for the initial generation from GSM8K. We use a different set of few-shot demonstrations for each dataset. Refer to our code for details.

```
[
  {
    'role': 'user',
    'content': 'There are 15 trees in the grove. Grove workers will plant trees in the grove
    ↪  today. After they are done, there will be 21 trees. How many trees did the grove
    ↪  workers plant today?'
  },
  {
    'role': 'assistant',
    'content': 'There were originally 15 trees in the grove.
      After the grove workers planted trees today, there are now 21 trees.
      So, the grove workers planted 21 - 15 = 6 trees today.
      Therefore, the answer (arabic numerals) is 6.'
  }
]
```

### G.2.2 INFORMAL TO FORMAL CONVERSION IN FORMAL PROOF

As explained in Section 3.3, we use `meta-llama/Llama-3.3-70B-Instruct` with few-shot prompting for converting the informal solutions from LLMs into the Isabelle format. Our conversion is based on the ideas in prior work (Wu et al., 2022; Zhou et al., 2024).

We employ LLM-based conversion, so the informal-to-formal conversion is not always perfect. We can remove syntactically invalid formal proofs using Isabelle, but it is difficult to remove unfaithful conversion, which results in formal proofs that are semantically different from the original informal solutions. However, this unfaithful conversion does not negatively influence the quality of our training data because we train PRMs on the task of detecting mistakes in the formal proofs using step-level error labels annotated using Isabelle, and we do not use the informal solutions during training. We expect that the converted formal proofs are often semantically identical or similar to the original informal solutions, but the converted formal proofs do not need to be always identical to the informal solutions.

The following is an example of a few-shot demonstration for the conversion.

### Few-Shot Demonstration for Statement Conversion

```
{
  'role': 'user',
  'content': 'Your task is to convert the following informal statement into a formal
  ↪  statement in Isabelle 2022.
    * Make your formal statement consistent with the provided Informal Statement.
    * The final answer can be wrong, but your formal statement should be faithful to the
    ↪  informal statement and should not correct the mistakes in the informal statement.
    * You should not use new variables in "shows".
```

```
    * The "shows" part is expected to be formatted as "variable = number" (e.g., "x = 3").

    ** Informal Statement **
    Gloria wants to buy the $129,000 mountain cabin that her friend Alfonso is selling. She
    ↪  only has $150 in cash. She intends to raise the remaining amount by selling her
    ↪  mature trees for lumber. She has 20 cypress trees, 600 pine trees, and 24 maple
    ↪  trees. She will get $100 for each cypress tree, $300 for a maple tree, and $200 per
    ↪  pine tree. After paying Alfonso for the cabin, how much money will Gloria have left?'
},
{
  'role': 'assistant',
  'content': 'theorem example:
        (* Gloria wants to buy the $129,000 mountain cabin that her friend Alfonso is
        ↪  selling. *)
        assumes "(Cabin_cost::nat) = 129000"
        (* She only has $150 in cash. *)
            and "(Gloria_cash::nat) = 150"
        (* She intends to raise the remaining amount by selling her mature trees for lumber.
        ↪  *)
        (* She has 20 cypress trees, 600 pine trees, and 24 maple trees. *)
            and "(Cypress_count::nat) = 20"
            and "(Pine_count::nat) = 600"
            and "(Maple_count::nat) = 24"
        (* She will get $100 for each cypress tree, $300 for a maple tree, and $200 per pine
        ↪  tree. *)
            and "(Cypress_price::nat) = 100"
            and "(Maple_price::nat) = 300"
            and "(Pine_price::nat) = 200"
        (* After paying Alfonso for the cabin, how much money will Gloria have left? *)
            and "(After_paying_Alfonso::nat) = Gloria_cash + Cypress_count * Cypress_price +
            ↪  Maple_count * Maple_price + Pine_count * Pine_price - Cabin_cost"
        (* Final Answer -- The answer is 200. *)
        shows "After_paying_Alfonso = 200"'
}
```

Few-Shot Demonstration for Proof Conversion

```
{
  'role': 'user',
  'content': 'Your task is to convert the following informal proof into a formal proof in
  ↪  Isabelle 2022.
    * The input informal proof can be wrong, but your formal proof should be faithful to the
    ↪  informal proof and should not correct the mistakes in the informal proof.
    * In your formal proof, use variables defined in the provided Formal Statement.
    * Use sledgehammer.
    * You should use defined variables whenever possible and should not write equations that
    ↪  only contain numbers.
    * The last step is expected to be the same as the equation shown in the \shows" section
    ↪  of the Formal Statement.
    * Include informal statements and proof as comments.

    ** Informal Statement **
    Gloria wants to buy the $129,000 mountain cabin that her friend Alfonso is selling. She
    ↪  only has $150 in cash. She intends to raise the remaining amount by selling her
    ↪  mature trees for lumber. She has 20 cypress trees, 600 pine trees, and 24 maple
    ↪  trees. She will get $100 for each cypress tree, $300 for a maple tree, and $200 per
    ↪  pine tree. After paying Alfonso for the cabin, how much money will Gloria have left?
    Final Answer -- The answer is 200.

    ** Informal Proof **
    Gloria needs 129,000 dollars for the cabin, but she only has 150 dollars.
    So she needs 129,000 - 150 = 128,850 dollars.
    She will get 100 dollars for each of the 20 cypress trees, which is 20 x 100 = 2,000
    ↪  dollars.
    She will get 300 dollars for each of the 24 maple trees, which is 24 x 300 = 7,200
    ↪  dollars.
    She will get 200 dollars for each of the 600 pine trees, which is 600 x 200 = 120,000
    ↪  dollars.
    In total, she will get 2,000 + 7,200 + 120,000 = 129,200 dollars from selling the trees.
    After selling the trees, she will have 129,200 dollars.
    She will pay 129,000 dollars for the cabin, so she will have 129,200 - 129,000 = 200
    ↪  dollars left.
    Therefore, the answer (arabic numerals) is 200.

    ** Formal Statement **
    theorem example:
        (* Gloria wants to buy the $129,000 mountain cabin that her friend Alfonso is
        ↪  selling. *)
        assumes "(Cabin_cost::nat) = 129000"
```

```
        (* She only has $150 in cash. *)
            and "(Gloria_cash::nat) = 150"
        (* She intends to raise the remaining amount by selling her mature trees for lumber.
        ↪  *)
        (* She has 20 cypress trees, 600 pine trees, and 24 maple trees. *)
            and "(Cypress_count::nat) = 20"
            and "(Pine_count::nat) = 600"
            and "(Maple_count::nat) = 24"
        (* She will get $100 for each cypress tree, $300 for a maple tree, and $200 per pine
        ↪  tree. *)
            and "(Cypress_price::nat) = 100"
            and "(Maple_price::nat) = 300"
            and "(Pine_price::nat) = 200"
        (* After paying Alfonso for the cabin, how much money will Gloria have left? *)
            and "(After_paying_Alfonso::nat) = Gloria_cash + Cypress_count * Cypress_price +
            ↪  Maple_count * Maple_price + Pine_count * Pine_price - Cabin_cost"
        (* Final Answer -- The answer is 200. *)
        shows "After_paying_Alfonso = 200"'
    },
  {
    'role': 'assistant',
    'content': 'theorem example:
        (* Gloria wants to buy the $129,000 mountain cabin that her friend Alfonso is
        ↪  selling. *)
        assumes "(Cabin_cost::nat) = 129000"
        (* She only has $150 in cash. *)
            and "(Gloria_cash::nat) = 150"
        (* She intends to raise the remaining amount by selling her mature trees for lumber.
        ↪  *)
        (* She has 20 cypress trees, 600 pine trees, and 24 maple trees. *)
            and "(Cypress_count::nat) = 20"
            and "(Pine_count::nat) = 600"
            and "(Maple_count::nat) = 24"
        (* She will get $100 for each cypress tree, $300 for a maple tree, and $200 per pine
        ↪  tree. *)
            and "(Cypress_price::nat) = 100"
            and "(Maple_price::nat) = 300"
            and "(Pine_price::nat) = 200"
        (* After paying Alfonso for the cabin, how much money will Gloria have left? *)
            and "(After_paying_Alfonso::nat) = Gloria_cash + Cypress_count * Cypress_price +
            ↪  Maple_count * Maple_price + Pine_count * Pine_price - Cabin_cost"
        (* Final Answer -- The answer is 200. *)
        shows "After_paying_Alfonso = 200"
    proof -
        (* Gloria needs 129,000 dollars for the cabin, but she only has 150 dollars. *)
        (* So she needs 129,000 - 150 = 128,850 dollars. *)
        have "Cabin_cost - Gloria_cash = 128850"
            sledgehammer
        (* She will get 100 dollars for each of the 20 cypress trees, which is 20 × 100 =
        ↪  2,000 dollars. *)
        then have "Cypress_count * Cypress_price = 2000"
            sledgehammer
        (* She will get 300 dollars for each of the 24 maple trees, which is 24 × 300 = 7,200
        ↪  dollars. *)
        then have "Maple_count * Maple_price = 7200"
            sledgehammer
        (* She will get 200 dollars for each of the 600 pine trees, which is 600 × 200 =
        ↪  120,000 dollars. *)
        then have "Pine_count * Pine_price = 120000"
            sledgehammer
        (* In total, she will get 2,000 + 7,200 + 120,000 = 129,200 dollars from selling the
        ↪  trees. *)
        then have "Cypress_count * Cypress_price + Maple_count * Maple_price + Pine_count *
        ↪  Pine_price = 129200"
            sledgehammer
        (* After selling the trees, she will have 129,200 dollars. *)
        (* She will pay 129,000 dollars for the cabin, so she will have 129,200 - 129,000 =
        ↪  200 dollars left. *)
        then have "(Cypress_count * Cypress_price + Maple_count * Maple_price + Pine_count *
        ↪  Pine_price) - Cabin_cost = 200"
            sledgehammer
        (* Therefore, the answer (arabic numerals) is 200. *)
        then have "After_paying_Alfonso = 200"
            sledgehammer
        show ?thesis
            sledgehammer
    qed'
  }
```

### G.2.3 Automatic Step-level Proof Verification in Isabelle

Second, we automatically annotate step-level error labels using Isabelle. Isabelle is designed for solution-level verification, so we need to write wrapper code to use it for step-level verification. We also explain how we used the Sledgehammer function in Isabelle.

**Lemmas and Sledgehammer in Isabelle.** In Isabelle, we need to manually specify lemmas and theorems to construct proofs. For example, in the following toy example, we need to manually specify `rule add_assoc` to properly prove the theorem.

```
theory Addition_Assoc_Manual
  imports Main
begin

lemma add_assoc_manual: "(a + b) + c = a + (b + c)"
proof -
  show ?thesis
    by (rule add_assoc)
qed

end
```

We can instruct the conversion LLMs to generate proper lemmas, but this makes the conversion LLMs make more syntax errors. To make the conversion easier, we make use of Isabelle's Sledgehammer tool (Paulson & Blanchette, 2012). Sledgehammer automates the process of specifying lemmas by invoking external automated theorem provers (ATPs) to suggest proof steps or entire proofs. It translates the current proof state into a form suitable for ATPs, runs them, and then attempts to reconstruct the suggested proof within Isabelle's logical framework. This significantly reduces the manual effort required, improves productivity, and bridges the gap between interactive and automated proving. Using Sledgehammer, we can simplify the above theorem as follows to verify it using Isabelle. We use a wrapper for Isabelle (Welleck, 2023) for using Sledgehammer.

```
theory Addition_Assoc_Sledgehammer
  imports Main
begin

lemma add_assoc_manual: "(a + b) + c = a + (b + c)"
proof -
  show ?thesis
    sledgehammer
qed

end
```

Nevertheless, it is not guaranteed that Sledgehammer can always find a lemma when the statement is correct. In other words, verification using Sledgehammer does not cause false negatives (i.e., overlook mistakes in proofs), but it can false-positively classify a correct step as a mistake when we use lemmas generated by Sledgehammer. In this work, we focused on GSM8K-level questions, for which Isabelle's Sledgehammer can often reliably provide correct lemmas when they exist. Therefore, we annotate steps as incorrect when Sledgehammer fails to find the lemma, and this process yields reasonably reliable training data. However, we need to be careful when we use Sledgehammer with more complex tasks.

For future work, we can extend our framework to more complex mathematical reasoning tasks, without using Sledgehammer. In complex tasks, even when Sledgehammer fails to find the lemma, it does not mean the statement is incorrect. Therefore, we cannot reliably annotate whether a statement is correct. In this case, one possible approach is to use a training task of verifying whether a pair of a statement and a lemma is correct. Specifically, each step of initial responses consists of a pair of a statement and a lemma like "(a + b) + c = a + (b + c) by (rule add_assoc)", which can be reliably verified by Isabelle.

**Format verification.** As we use LLMs for the conversion, it is possible that the conversion generates proofs in an invalid format. To filter out theorems in the invalid format, we use the "sorry" keyword of Isabelle. The "sorry" keyword serves as a placeholder for incomplete or unproven proofs, allowing the theorem to be accepted by the system without a formal justification. By inserting "sorry" into all generated proof steps, we can isolate and verify only the syntactic correctness theorems.

For example, if the following theorem gets errors from Isabelle, we can detect syntax errors and filter this theorem out from our dataset. In this case, the expression `babysitting_minutes ×` `(Weng_hourly_wage / 60)` contains the symbol ×, which is not a valid multiplication operator in Isabelle syntax.

```
theorem example:
    assumes "(Weng_hourly_wage::real) = 12"
        and "(babysitting_minutes::real) = 50"
        and "(babysitting_hours::real) = babysitting_minutes / 60"
        and "(Weng_earnings::real) = Weng_hourly_wage * babysitting_hours"
    shows "Weng_earnings = 10"
proof -
    have "Weng_hourly_wage / 60 = 0.20"
        sorry
    then have "babysitting_minutes × (Weng_hourly_wage / 60) = 10"
        sorry
    then have "Weng_earnings = 10"
        sorry
    thus ?thesis
        sorry
qed
```

For this input, Isabelle returns the following error.

```
Step error: Inner syntax error (line 1)\nat \"? ( Weng_hourly_wage / 60 ) = 10\"\nFailed to
↪  parse prop\nAt command \"have\" (line 1)
```

**Step-level verification.** By default, Isabelle halts at the first encountered error and does not provide a step-by-step verification of a proof. To enable independent verification of each step in a multi-step proof, we insert the "sorry" keyword in all but one step. This allows Isabelle to type-check and parse each step individually, even if other steps are incomplete or invalid.

The following example is for verifying the third step independently. For each theorem, we run Isabelle once per step to isolate and validate its correctness.

```
theorem example:
    assumes "(wallet_cost::nat) = 100"
        and "(betty_savings::nat) = wallet_cost div 2"
        and "(parent_contribution::nat) = 15"
        and "(grandparent_contribution::nat) = 2 * parent_contribution"
        and "(total_savings::nat) = betty_savings + parent_contribution +
        ↪  grandparent_contribution"
        and "(additional_needed::nat) = wallet_cost - total_savings"
    shows "additional_needed = 5"
proof -
    have "betty_savings = wallet_cost div 2"
        sorry
    then have "betty_savings = 50"
        sorry
    have "grandparent_contribution = 2 * parent_contribution"
        sledgehammer
    then have "grandparent_contribution = 30"
        sorry
    then have "parent_contribution + grandparent_contribution = 45"
        sorry
    then have "total_savings = 95"
        sorry
    then have "additional_needed = wallet_cost - total_savings"
        sorry
    then have "additional_needed = 5"
        sorry
    thus ?thesis
        sorry
```

## H  INPUT FORMAT AND POSTPROCESSING FOR LLM-BASED PRMS

This section provides details of the input format and postprocessing of LLM-based PRMs.

### H.1  FOVER-PRMS AND THE BASELINE PRMS

First, we describe the input format for FOVER-PRMs and the baseline LLM-based PRMs, which are based on Llama 3.1 8B and Qwen 2.5 7B. FOVER-PRMs are trained on FOVER-80K, so the input format has the same format as the training data. The only difference is that we replace all step-level labels with "correct" in the input. This preprocessing allows us to provide the whole input once to get the step-level predictions for all steps. The following is an example input for GSM8K.

```
[
  {
    'role': 'user',
    'content': '** Problem **
    Alice is 7 years older than Beth, who is 5 years older than Erica. What is the
    ↪  difference between the ages of Alice and Erica, if Erica is 30 years old?

    ** Task **
    Your task is to evaluate the accuracy of each step in the provided solution to the
    ↪  above question. For each step, respond with "correct" if the reasoning is logically
    ↪  valid and mathematically sound, or if the step is a general statement or transition
    ↪  that does not contain reasoning. Respond with "incorrect" if the step includes any
    ↪  errors or flawed logic.

    ** Sotluion **
    Since Erica is 30 years old, and Beth is 5 years older than Erica, Beth is 30 + 5 = 35
    ↪  years old.'
  },
  {'role': 'assistant', 'content': 'correct'},
  {'role': 'user', 'content': 'Alice is 7 years older than Beth, who is 35 years old. '},
  {'role': 'assistant', 'content': 'correct'},
  {'role': 'user', 'content': 'So, Alice is 35 + 7 = 42 years old.'},
  {'role': 'assistant', 'content': 'correct'},
  {'role': 'user', 'content': "The difference between Alice's age and Erica's age is 42 – 30
  ↪  = 12 years."},
  {'role': 'assistant', 'content': 'correct'},
  {'role': 'user', 'content': 'Therefore, the answer (arabic numerals) is 12.'},
  {'role': 'assistant', 'content': 'correct'}
]
```

Next, we describe the postprocessing for FOVER-PRMs and the baseline LLM-based PRMs.

**Extracting logits.** Since we use causal LLMs as PRMs, we extract the model's predictions for the tokens immediately preceding the dummy step-level labels (e.g., "correct") in the input.

**Computing step-level scores.** At each identified position, we extract the logits corresponding to the tokens "correct" and "incorrect". We then apply the softmax function over these two logits to compute the probability assigned to "correct". This probability serves as the step-level score.

### H.2  EXISTING PRMS

**PRMs based on Llama 3.1 8B.** In RLHFlow-Llama3.1-8B-DeepSeek and RLHFlow-Llama3.1-8B-Mistral (Xiong et al., 2024), the input format is mostly similar to ours, with the key difference being the use of "+" and "-" instead of "correct" and "incorrect".[3] For these models, we apply our input format and postprocessing procedures with a simple substitution of "correct" with "+".

**PRMs based on Qwen 2.5 7B.** Qwen2.5-Math-7B-PRM800K (Zheng et al., 2025) and Qwen2.5-Math-PRM-7B (Zhang et al., 2025) are supported by vLLM (Kwon et al., 2023). We follow the input format specified in their respective model descriptions and adopt the reward modeling in vLLM.[4] For Qwen2.5-7B-Skywork-PRM (He et al., 2024), we use a code provided by the authors.[5]

---

[3] https://github.com/RLHFlow/RLHF-Reward-Modeling/tree/main/math-rm
[4] https://docs.vllm.ai/en/latest/models/pooling_models.html
[5] https://github.com/SkyworkAI/skywork-o1-prm-inference

# I  TRAINING SETTINGS

This section provides details of the training settings for our FOVER-PRMs.

**Training Data Format** FOVER-80K includes balanced step-level error labels of 50% of "correct" and "incorrect" labels. To balance the step-level error labels, we set `mask_history: True` in LLaMA-Factory (Zheng et al., 2024), which configures the model to use only the last step of each conversation during training. We postprocess the training data by truncating reasoning steps so that the final steps are balanced between "correct" and "incorrect" labels. This process is functionally equivalent to selectively masking certain steps to achieve balanced labels, but is easier to implement within LLaMA-Factory.

For example, from the case in Appendix F.2, we can make a training case for the "incorrect" label. In each instance, models are only trained on the prediction for a single token, "correct" or "incorrect" in the last step, and the other tokens are masked.

```
{
  'id': 'bigmath_orca_math_138943_Llama-3.1-8B-Instruct',
  'error_labels': [True, False, True, True, True],
  'messages': [
  {
    'role': 'user',
    'content': '** Problem **
      Generate a proof for the following theorem in the Isabelle proof assistant format.

      theorem example:
        assumes "(Stella_Artois_speed::real) = 150"
            and "(total_distance::real) = 500"
            and "(total_time::real) = 1.694915254237288"
            and "(first_car_speed::real) = (total_distance - Stella_Artois_speed *
            ↪  total_time) / total_time"
        shows "first_car_speed = 145.2"

      ** Task **
      Your task is to evaluate the accuracy of each step in the provided solution to the
      ↪  above question. For each step, respond with "correct" if the reasoning is logically
      ↪  valid and mathematically sound, or if the step is a general statement or transition
      ↪  that does not contain reasoning. Respond with "incorrect" if the step includes any
      ↪  errors or flawed logic.

      ** Sotluion **
      have "Stella_Artois_speed * total_time = 150 * total_time"'
  },
  {'role': 'assistant', 'content': 'correct'},
  {
    'role': 'user',
    'content': 'then have "Stella_Artois_speed * total_time = 254.1794318175"
      then have "total_distance - Stella_Artois_speed * total_time = 245.8205681825"'
  },
  {'role': 'assistant', 'content': 'incorrect'},
  ]
```

**Training Parameters** We fine-tune all model parameters and do not use parameter-efficient techniques. We use the AdamW optimizer (Loshchilov & Hutter, 2019) and select the learning rate based on the average Best-of-K performance on the validation tasks: Orca-Math (Mitra et al., 2024) and two tasks in BBH (Suzgun et al., 2023) (Logical Deduction (three objects) and Boolean Expressions). We evaluate models trained with the learning rate 1e-6, 2e-6, 5e-6, and 1e-5, and select the model with the best average performance on the validation tasks.

We use the parameters in Table 9 in all models, and we did not conduct hyperparameter tuning for these parameters. Please refer to the configuration files in our code for further details.

Table 9: Hyperparameters in training on FOVER

| Parameter | Value |
|---|---|
| Number of Epochs | 1 |
| Batch size | 32 |
| Learning Rate Warm up and Decay Strategy | Linear |
| Learning Rate Warm up Ratio | 0.5 |

## J  EVALUATION SETTINGS OF BEST-OF-K

This section provides details of the evaluation settings for Best-of-K in Section 4.

**Benchmarks.** Table 10 shows sources of reasoning benchmarks evaluated in Best-of-K in Section 4.

Table 10: Reasoning benchmarks evaluated in the Best-of-K experiments.

| Dataset | Source |
|---|---|
| GSM8K | https://huggingface.co/datasets/openai/gsm8k |
| MATH | https://github.com/hendrycks/math |
| AQuA-RAT | https://huggingface.co/datasets/deepmind/aqua_rat |
| AIME (2016-2024) | https://huggingface.co/datasets/di-zhang-fdu/AIME_1983_2024 |
| FOLIO | https://huggingface.co/datasets/yale-nlp/FOLIO |
| LogicNLI | https://huggingface.co/datasets/tasksource/LogicNLI |
| ANLI | https://huggingface.co/datasets/facebook/anli |
| HANS | https://github.com/tommccoy1/hans |
| MMLU | https://huggingface.co/datasets/sam-paech/mmlu-pro-nomath-sml |
| BBH | https://github.com/suzgunmirac/BIG-Bench-Hard/tree/main/bbh |

**Initial generation prompts.** Table 11 shows detailed settings of generating $K = 5$ responses for the Best-of-K evaluation in Section 4.2. We create new few-shot examples or modify few-shot demonstrations used in prior work to enhance the quality and to simplify the post-processing procedure. For example, we add line breaks between reasoning steps in all tasks. An example prompt for GSM8K is provided in Appendix G.1.1. Please also refer to our code for further details.

Table 11: Detailed settings for Best-of-K downstream evaluation

| Dataset | Few-shot Examples for Initial Generation | Answer Matching |
|---|---|---|
| GSM8K | (Kojima et al., 2022) [6] | Exact match after extraction and conversion to integer |
| MATH | (Lewkowycz et al., 2022, Appendix D.2) | (Lewkowycz et al., 2022, Appendix G) |
| AQuA-RAT | Made by us (3-shot) | Exact match after extraction |
| AIME | Made by us (3-shot) | Exact match after extraction and conversion to integer |
| FOLIO | Made by us (2-shot) | Exact match after extraction |
| LogicNLI | Made by us (3-shot) | Exact match after extraction |
| ANLI | Made by us (3-shot) | Exact match after extraction |
| HANS | Made by us (2-shot) | Exact match after extraction |
| MMLU | Made by us (4-shot) | Exact match after extraction |
| BBH | (Suzgun et al., 2023)[7] | Exact match after extraction |

---

[6] https://github.com/kojima-takeshi188/zero_shot_cot/blob/
5ef330fcdeec0cd26aee27943504f91f8ec1c33c/utils.py#L328

[7] https://github.com/suzgunmirac/BIG-Bench-Hard/blob/main/cot-prompts

# K  ADDITIONAL RESULTS

This section provides additional experimental results.

## K.1  ABLATION STUDY OF TRAINING TASKS OF FOVER-80K

We evaluate PRMs trained on three variants (subsets) of FOVER-80K: formal logic only, formal theorem proving only, and a combination of both (i.e., the original FOVER-80K). Table 12 and 13 show results of Best-of-K on 12 reasoning datasets and step-level verification in ProcessBench. We observe that both formal logic and theorem proving tasks independently yield statistically significant improvements over the baseline PRMs, verifying that both of these tasks are effective at improving PRMs on broad reasoning tasks.

Table 12: Ablation study of PRMs trained on different versions of FOVER-80K in Best-of-K (K=5). *: Statistically significant improvement over the baseline ($p < 0.05$, paired bootstrap).

| Training Dataset | | Math | | | | Logic | | NLI | | MMLU | BBH | | | Average |
|---|---|---|---|---|---|---|---|---|---|---|---|---|---|---|
| Formal Logic | Formal Proof | GSM8K | MATH | AQuA | AIME | FOLIO | LogicNLI | ANLI | HANS | Pro-NoMath | Temporal | Tracking | Sorting | |
| **PRMs based on Llama 3.1 8B select the best response from $K = 5$ responses generated by Llama 3.1 8B** | | | | | | | | | | | | | | |
| ✓ | – | 88.4 | 40.8 | 69.7* | 4.4 | 65.5* | 43.6 | 30.8* | 83.6* | 55.2 | 89.2 | 96.0* | 40.8 | 59.0* |
| – | ✓ | 89.2 | 41.2 | 70.1* | 2.0 | 61.1 | 39.2 | 28.8 | 77.2 | 55.6 | 97.2* | 94.0* | 40.0 | 58.0* |
| ✓ | ✓ | 86.4 | 43.2 | 65.7 | 4.0 | 64.0* | 44.8* | 28.8 | 82.8* | 57.2 | 97.6* | 93.2* | 38.4 | 58.8* |
| **PRMs based on Qwen 2.5 7B select the best response from $K = 5$ responses generated by Qwen 2.5 7B** | | | | | | | | | | | | | | |
| ✓ | – | 92.8* | 65.2 | 81.1 | 10.8 | 61.6 | 46.8 | 31.2 | 87.2* | 61.2 | 90.8 | 89.2 | 32.8* | 62.6* |
| – | ✓ | 91.6 | 66.0 | 81.5 | 12.4 | 62.1 | 50.4 | 32.0 | 84.0 | 60.4 | 91.6 | 90.8 | 30.4 | 62.8* |
| ✓ | ✓ | 92.8* | 66.4 | 82.3 | 13.2 | 65.5 | 48.8 | 32.4 | 85.2 | 60.8 | 91.6 | 89.6 | 31.2 | 63.3* |

Table 13: Ablation study of FOVER-PRMs on step-level verification on ProcessBench (AUROC). *: Statistically significant improvement over the baseline PRMs ($p < 0.05$, paired bootstrap).

| | (a) Llama 3.1 as PRMs | | | | | | | (b) Qwen 2.5 as PRMs | | | | | |
|---|---|---|---|---|---|---|---|---|---|---|---|---|---|

| Logic | Proof | GSM8K | MATH | Olympiad | Omni | Average | Logic | Proof | GSM8K | MATH | Olympiad | Omni | Average |
|---|---|---|---|---|---|---|---|---|---|---|---|---|---|
| ✓ | – | 79.2* | 79.8* | 81.2* | 76.5* | 79.2* | ✓ | – | 84.9* | 83.3* | 83.5* | 82.5* | 83.5* |
| – | ✓ | 87.9* | 82.5* | 82.7* | 80.9* | 83.5* | – | ✓ | 90.5* | 87.9* | 84.3* | 84.8* | 86.9* |
| ✓ | ✓ | 80.0* | 74.1* | 74.8* | 74.7* | 75.9* | ✓ | ✓ | 90.8* | 89.1* | 84.6* | 86.0* | 87.6* |

## K.2 Ablation Study of Training Hyperparameters

We analyze training hyperparameters that may influence cross-task generalization of FOVER. We train PRMs on FOVER-80K with different learning rates, label distributions, and dataset sizes, and evaluate performance on the validation tasks: Orca-Math (Mitra et al., 2024) and two tasks in BBH (Suzgun et al., 2023) (Logical Deduction (three objects) and Boolean Expressions).

In summary, we observe that training on FOVER-80K is reasonably robust to these training hyper-parameters. Since the influence of the learning rate on the final performance is relatively large, we perform hyperparameter search only over the learning rate using the validation tasks for FOVER-PRMs. We use the label distribution of 0.5 and use the dataset size of 40k in our experiments.

**Learning rate.** We evaluate PRMs trained on FOVER-80K using different learning rates: 1e-6, 2e-6, 5e-6, and 1e-5. Figure 7a reports the average accuracy on the validation tasks for PRMs trained on the 40k-instance training set. We observe that the learning rate has a relatively substantial impact on the transferability and that the optimal learning rate varies across models and evaluation tasks.

**Label distribution.** We evaluate PRMs trained on FOVER with varying proportions of correct and incorrect step-level labels. Specifically, we construct variants of training datasets that include both formal logic and proof tasks, with 25%, 50%, and 75% correct labels, each containing 20k instances. Figure 7b reports the average accuracy on the validation tasks. The influence of the label distribution is relatively small, and we do not observe a consistent trend across label distributions; different models and evaluation datasets exhibit varying behavior.

**Dataset size.** We evaluate PRMs trained on different sizes of FOVER-80K, which includes both formal logic and proof tasks. Specifically, we train models on 5k, 10k, 20k, and 40k instances. To isolate the effect of training data diversity, all models are trained for the same number of steps. Figure 7c presents the accuracy on the validation tasks, showing that larger and more diverse training data often improves cross-task generalization.

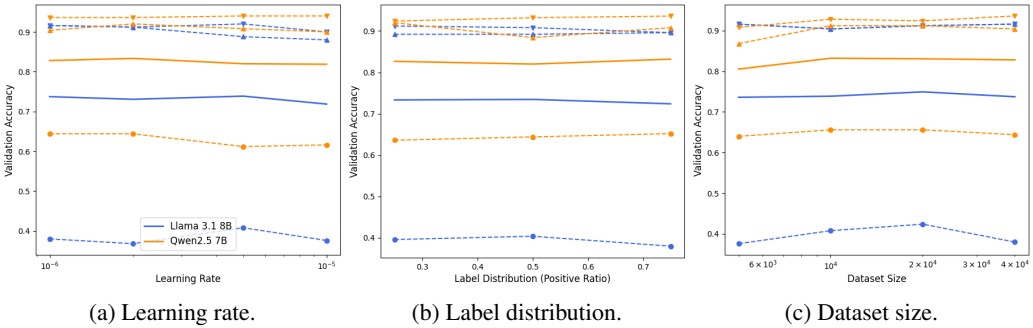

(a) Learning rate.  (b) Label distribution.  (c) Dataset size.

Figure 7: Influence of hyperparameters on the cross-task generalization of PRMs trained on FOVER. The thick lines represent the average accuracy in the validation tasks. The dashed lines represent performance on validation tasks: ● Orca-Math, ▲ Logical Deduction, and ▼ Boolean Expressions.

### K.3 INFERENCE-TIME SCALING OF FOVER-PRMS

In Section 4.2, we evaluate Best-of-K performance with $K = 5$. To show that our PRMs exhibit inference-time scaling, we evaluate performance with larger $K$. Table 14 shows the performance of PRMs based on Llama 3.1 8B with larger $K$ up to $K = 30$ on four tasks. In this table, to more effectively leverage a larger number of responses, we selected the final answer using a weighted majority vote for the final answer, where the weight for each solution is determined by instance-level score assigned by PRMs.

FOVER-PRM based on Llama 3.1 8B exhibits inference-time scaling, and FOVER-PRM consistently outperforms the original model, except on MATH.

Table 14: Inference-scaling of FOVER-PRMs based on Llama 3.1 8B in Best-of-K performance with weighted majority vote

| K | MATH | | FOLIO | | HANS | | Tracking | |
|---|---|---|---|---|---|---|---|---|
| | Original | FOVER | Original | FOVER | Original | FOVER | Original | FOVER |
| 5 | **46.0** | 45.6 | 59.1 | **64.5** | 73.6 | **80.0** | 94.0 | **94.8** |
| 10 | **46.8** | 46.0 | 61.6 | **63.1** | 73.6 | **80.8** | 94.8 | **96.0** |
| 15 | **49.2** | 47.6 | 64.5 | **67.0** | 73.6 | **80.4** | 94.8 | **95.6** |
| 30 | **49.2** | **49.2** | 65.5 | **68.5** | 74.0 | **80.0** | 94.8 | **96.0** |

## K.4 PRMs Trained on a Combination of FoVer-80K and PRM800K

This section provides details of the experiment in Section 4.4 showing the performance of PRMs trained on a combination of FoVer-80K and PRM800K.

**Settings.** We use PRM800K (Lightman et al., 2024), a popular and one of the few publicly available training datasets for PRMs. Similar to FoVer-80K, we preprocess it and make a dataset of 20K steps with balanced step-labels.

**Results.** Table 15 reports the Best-of-K performance of PRMs trained on FoVer-80K and PRM800K across 12 benchmarks. First, before combining the two datasets, we observe that PRMs trained on PRM800K perform worse than PRMs trained on FoVer-80K on nearly all tasks. However, combining FoVer-80K and PRM800K generally yields better or competitive performance compared to PRMs trained solely on FoVer-80K. Notably, PRMs trained in the combined dataset outperform FoVer-PRM on GSM8K, MATH, AQuA, and the tracking task of BBH. In short, PRM800K, which consists of mathematical reasoning tasks, enhances the verification capabilities of PRMs in mathematics when combined with FoVer. Although PRMs trained on the combined dataset show degraded performance compared to FoVer-PRMs on a few tasks, including FOLIO and LogicNLI for Llama 3.1 8B and MMLU for Qwen 2.5 7B, these results indicate that integrating FoVer with other datasets can improve their performance or alter PRM properties.

Table 15: Best-of-K (K=5) performance of PRMs trained on FoVer-80K and PRM800K.

(a) PRMs based on Llama 3.1 8B select the best response from $K = 5$ responses generated by Llama 3.1 8B

| Training Data | Math | | | | Logic | | NLI | | MMLU | BBH | | | Average |
|---|---|---|---|---|---|---|---|---|---|---|---|---|---|
| | GSM8K | MATH | AQuA | AIME | FOLIO | LogicNLI | ANLI | HANS | Pro-NoMath | Temporal | Tracking | Sorting | |
| FoVer-80K | 86.4 | 43.2 | 65.7 | 4.0 | 64.0 | 44.8 | 28.8 | 82.8 | 57.2 | 97.6 | 93.2 | 38.4 | 58.8 |
| PRM800K | 89.2 | 41.2 | 65.0 | 2.4 | 58.6 | 39.2 | 32.0 | 80.0 | 56.8 | 89.6 | 91.2 | 37.2 | 56.9 |
| FoVer-80K + PRM800K | 90.0 | 44.8 | 68.5 | 3.6 | 60.1 | 40.0 | 31.6 | 83.2 | 57.2 | 96.8 | 95.2 | 36.8 | 59.0 |

(b) PRMs based on Qwen 2.5 7B select the best response from $K = 5$ responses generated by Qwen 2.5 7B

| Training Data | Math | | | | Logic | | NLI | | MMLU | BBH | | | Average |
|---|---|---|---|---|---|---|---|---|---|---|---|---|---|
| | GSM8K | MATH | AQuA | AIME | FOLIO | LogicNLI | ANLI | HANS | Pro-NoMath | Temporal | Tracking | Sorting | |
| FoVer-80K | 92.8 | 66.4 | 82.3 | 13.2 | 65.5 | 48.8 | 32.4 | 85.2 | 60.8 | 91.6 | 89.6 | 31.2 | 63.3 |
| PRM800K | 92.8 | 64.8 | 81.5 | 13.2 | 62.6 | 48.4 | 32.4 | 85.2 | 60.4 | 90.0 | 89.6 | 32.2 | 62.8 |
| FoVer-80K + PRM800K | 93.6 | 66.8 | 83.1 | 13.2 | 62.6 | 49.2 | 32.4 | 85.6 | 58.0 | 91.6 | 89.6 | 32.2 | 63.2 |

## K.5 REFERENCE METHODS FOR BEST-OF-K: MAJORITY-OF-K AND PASS@K.

Table 16 shows the performance of reference methods on datasets evaluated in Section 4.2.

**Majority@K.** Majority@K (or self-consistency) (Wang et al., 2023), which takes the majority vote of the final answers. It is known to be a strong baseline (Zhang et al., 2025) and is competitive with the Best-of-K performance of FOVER-PRMs.

**Oracle Verification.** Oracle Verification (or Pass@K) is the performance when we always select the solution with the correct final answer if it exists, which is the upper-bound of Best-of-K.

Table 16: Performance of reference methods on datasets in Table 3.

(a) PRMs based on Llama 3.1 8B select the best response from $K = 5$ responses generated by Llama 3.1 8B

| Methods | Math | | | | Logic | | NLI | | MMLU | BBH | | | Average |
|---|---|---|---|---|---|---|---|---|---|---|---|---|---|
| | GSM8K | MATH | AQuA | AIME | FOLIO | LogicNLI | ANLI | HANS | Pro-NoMath | Temporal | Tracking | Sorting | |
| FOVER-PRM | 86.4 | 43.2 | 65.7 | 4.0 | 64.0 | 44.8 | 28.8 | 82.8 | 57.2 | 97.6 | 93.2 | 38.4 | 58.8 |
| Majority@K | 89.2 | 44.4 | 66.1 | 4.8 | 62.6 | 46.0 | 33.2 | 76.4 | 58.8 | 92.8 | 94.0 | 42.4 | 59.2 |
| Oracle Verification | 97.2 | 58.4 | 83.9 | 8.8 | 90.1 | 83.6 | 54.4 | 94.4 | 75.2 | 100.0 | 98.4 | 60.0 | 75.4 |

(b) PRMs based on Qwen 2.5 7B select the best response from $K = 5$ responses generated by Qwen 2.5 7B

| Methods | Math | | | | Logic | | NLI | | MMLU | BBH | | | Average |
|---|---|---|---|---|---|---|---|---|---|---|---|---|---|
| | GSM8K | MATH | AQuA | AIME | FOLIO | LogicNLI | ANLI | HANS | Pro-NoMath | Temporal | Tracking | Sorting | |
| FOVER-PRM | 92.8 | 66.4 | 82.3 | 13.2 | 65.5 | 48.8 | 32.4 | 85.2 | 60.8 | 91.6 | 89.6 | 31.2 | 63.3 |
| Majority@K | 91.6 | 65.2 | 83.9 | 12.8 | 65.0 | 50.0 | 32.4 | 84.0 | 61.6 | 88.4 | 86.4 | 31.2 | 62.7 |
| Oracle Verification | 96.8 | 73.2 | 91.7 | 19.2 | 78.3 | 80.0 | 49.6 | 89.2 | 75.2 | 93.2 | 93.2 | 38.8 | 73.2 |

K.6    MANUAL ANALYSIS OF IMPROVEMENTS BY FOVER

We manually analyze step-level scores for cases where FOVER-PRMs improves or degrades re-ranking in Best-of-K from baseline PRMs (§4.2). As Best-of-K evaluates the final answers of selected solutions without directly evaluating step-level verification by PRMs, we conduct manual analysis to verify that FOVER-80K actually improves the step-level verification of PRMs. We analyze step-level scores in GSM8K, ANLI, and the temporal sequence and word sorting task in BBH.

We analyze cases where FOVER-PRMs improve or degrade re-ranking in Best-of-K with regard to the final answers from the baseline PRMs based on the original LLMs. We categorize the reasoning for the improvement and degradation of the final answers into the following categories:

- The baseline PRMs assign wrong step-level scores, although FOVER-PRMs assign reasonable step-level scores, which represent the genuine improvement by FOVER.

- The original questions are ambiguous or the ground-truth labels are incorrect.

- The selected solutions with the correct final answers include incorrect intermediate steps. PRMs are expected to assign low scores to those solutions, although they produce the correct final answers.

- Both PRMs assign incorrect step-level scores, with the baseline selecting the correct answer by chance.

- FOVER-PRMs assign wrong step-level scores, although the baseline PRMs assign reasonable step-level scores, which represent the genuine improvement by FOVER.

Only the first and last categories represent cases where FOVER-PRMs genuinely ● improve or ● degrade step-level verification performance compared to the baseline PRMs.

Figure 8 shows the results of the manual analysis. Our manual analysis shows that FOVER-80K often genuinely ● improve step-level verification and rarely leads to ● degradation compared to the baseline PRMs. Among cases where FOVER-80K improved final answers in Best-of-K, we observe that FOVER-80K often improves the step-level verification, particularly in GSM8K and the temporal sequence task, where over 50% of cases are improved. Among cases where FOVER-80K degrades the final answers in Best-of-K, we observe that FOVER-80K degrades the step-level scores in less than 20% of the cases on most of the tasks.

Our analysis shows that much of the degradation in the final answers does not represent a decline in step-level verification. For example, there are many cases where solutions selected by the baselines included incorrect steps, although they produced the correct final answers by chance. When intermediate steps are wrong, PRMs should reject such solutions even if the final answers are correct, so this case does not represent degradation by FOVER.

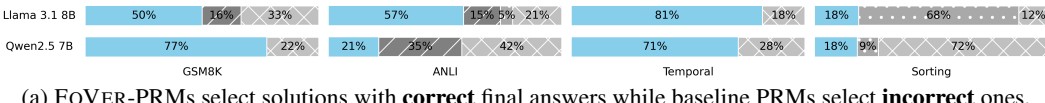

(a) FOVER-PRMs select solutions with **correct** final answers while baseline PRMs select **incorrect** ones.

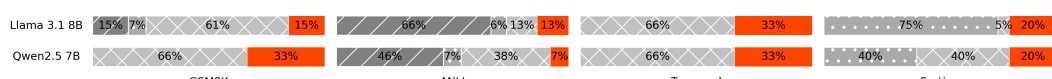

(b) FOVER-PRMs select solutions with **incorrect** final answers while baseline PRMs select **correct** ones.

Figure 8: Manual analysis of step-level scores in cases where FOVER-PRMs improve (8a) or degrade (8b) re-ranking in Best-of-K with regard to the final answers. The blue and red areas represent the cases in which FOVER genuinely ● improves and ● degrades the step-level verification scores. The other cases do not represent genuine improvement or degradation. ◌: The original questions are ambiguous or the ground-truth labels are incorrect. ⊕: The selected solutions with the correct final answers contain incorrect intermediate steps. ⊗: Both PRMs assign incorrect step-level scores.

We provide an example for each category.

● **The baseline PRMs assign wrong step-level scores, although FOVER-PRMs assign reasonable step-level scores.** The following example shows a response from Qwen 2.5 7B on ANLI. In this example, Step 1 of the above solution is wrong, which is correctly identified by FOVER-PRMs while the baseline PRM wrongly put a high score. In addition, in the second solution, the baseline PRM puts a low score to Step 2, which is correct. This case shows an example FOVER improves step-level verification.

---

**Problem:**

Premise: Cooper Manning SWAG-JACKED Don 'Magic' Juan??? Call the fashion police ... 'cause Cooper Manning straight-up STOLE his Super Bowl swag from Snoop's spiritual adviser, Bishop Don ""Magic"" Juan! Peyton and Eli's older bro was cruisin' around Super Bowl media night this week in a money suit – the same kind of outfit we've seen The Bishop rockin' for years. Twinsies.
Hypothesis: Payton and Eli Manning are both older than Cooper Manning.
Question: do the premises derive the hypothesis? Provide reasoning and answer with either ""entailment"", ""contradiction"" or ""neutral"".

**Ground truth:** neutral

---

| Step | Explanation | Baseline PRM | FOVER-PRM |
|------|-------------|--------------|-----------|
| **Solution selected by the baseline PRM (wrong final answer)** | | | |
| Step 0 | Premise states, "Peyton and Eli's older bro was cruisin' around Super Bowl media night this week in a money suit," implying that Cooper Manning is the brother of Peyton and Eli Manning. | 1.0000 | 0.9609 |
| Step 1 | Since Cooper Manning is described as the brother, it follows that Peyton and Eli Manning are older than Cooper Manning. | 1.0000 | 0.5625 |
| Step 2 | Therefore, the premise supports the hypothesis that Peyton and Eli Manning are both older than Cooper Manning. | 1.0000 | 0.9961 |
| Step 3 | The final answer is entailment. | 1.0000 | 1.0000 |
| Solution-level Score | | 1.0000 | 0.5625 |
| **Solution selected by FOVER-PRMs (correct final answer)** | | | |
| Step 0 | Premise states: "Peyton and Eli's older bro was cruisin' around Super Bowl media night this week in a money suit." This implies that Cooper Manning is the older brother of Peyton and Eli Manning. | 0.9922 | 0.9258 |
| Step 1 | Given that Cooper Manning is the older brother, it follows that Peyton and Eli Manning are both younger than Cooper Manning. | 1.0000 | 0.9922 |
| Step 2 | Therefore, the hypothesis "Peyton and Eli Manning are both older than Cooper Manning" is contradicted by the premise. | 0.0000 | 0.9844 |
| Step 3 | The final answer is contradiction. | 0.9531 | 0.9961 |
| Solution-level Score | | 0.0000 | 0.9258 |

✎ **The original questions are ambiguous or the ground-truth labels are incorrect.** The following example shows a response generated from Llama 3.1 8B on ANLI. In this case, the ground truth answer is "neutral". However, this problem is ambiguous, and both neutral and entailment are reasonable final answers. Therefore, this case does not represent a degradation by FOVER.

---

**Problem:**

Premise: Trying Tofu¡br¿Alicia was feeling a little adventurous. She wanted to try Tofu. She had never had it before. So she took a small bite. To her delight it was delicious.
Hypothesis: Tofu contains the letters to
Question: do the premises derive the hypothesis? Provide reasoning and answer with either ""entailment"", ""contradiction"" or ""neutral"".

**Ground truth:** neutral

---

| Step | Explanation | Baseline PRM | FOVER-PRM |
|------|-------------|:---:|:---:|
| **Solution selected by the baseline PRM (correct final answer)** | | | |
| Step 0 | The premise does not mention the letters in the word "tofu" at all. It talks about Alicia trying tofu and enjoying it. | 0.3496 | 0.7773 |
| Step 1 | The premise does not provide any information about the letters in the word "tofu." | 0.9922 | 0.9922 |
| Step 2 | Therefore, the final answer is neutral. | 0.9453 | 0.9766 |
| Solution-level Score | | 0.3496 | 0.7773 |
| **Solution selected by FOVER-PRMs (wrong final answer)** | | | |
| Step 0 | The premise mentions the word "tofu", which contains the letters "to". | 0.2813 | 0.9805 |
| Step 1 | The premise does not say anything about the taste, nutritional content, or any other properties of tofu, but it does provide evidence that the word "tofu" contains the letters "to". | 0.9922 | 0.9727 |
| Step 2 | Therefore, the final answer is entailment. | 0.5313 | 0.9570 |
| Solution-level Score | | 0.2813 | 0.9570 |

⊕ **Solutions selected by the baseline contain incorrect intermediate steps, although producing the correct final answers.** The following example shows a response generated from Llama 3.1 8B on the word sorting task in BBH. In the first solution, the final answer is correct; however, the first step includes `["farther" "insolent"]` in the sorted list, which is incorrect. When intermediate steps contain such errors, PRMs are expected to assign lower scores, potentially causing them to fail in selecting solutions with correct final answers. This behavior is expected and does not indicate that FOVER degrades verification performance.

---

**Problem:**

Sort the following words alphabetically: List: stephen amperage protocol crimea farther raillery tech insolent ping

**Ground truth:** amperage crimea farther insolent ping protocol raillery stephen tech

---

| Step | Explanation | Baseline PRM | FOVER-PRM |
|------|-------------|--------------|-----------|
| **Solution selected by the baseline PRM (correct final answer)** | | | |
| Step 0 | First, we look at the first letters of each word. "stephen": "s" (19th letter). "amperage": "a" (1st letter). "protocol": "p" (16th letter). "crimea": "c" (3rd letter). "farther": "f" (6th letter). "raillery": "r" (18th letter). "tech": "t" (20th letter). "insolent": "i" (9th letter). "ping": "p" (16th letter). We now have: ["amperage"] ¡ "crimea" ¡ ["farther" "insolent"] ¡ ["ping" "protocol"] ¡ "raillery" ¡ "stephen" ¡ "tech". | 0.5938 | 0.1641 |
| Step 1 | Now let's sort this subpart ["farther" "insolent"] by looking at their second letters. The second letter: "farther": "a" (1st letter). "insolent": "n" (14th letter). We now have: "farther" ¡ "insolent" ¡ ["ping" "protocol"] ¡ "raillery" ¡ "stephen" ¡ "tech". | 0.9883 | 0.6523 |
| Step 2 | Now let's sort this subpart ["ping" "protocol"] by looking at their second letters. The second letter: "ping": "i" (9th letter). "protocol": "r" (18th letter). We now have: "ping" ¡ "protocol" ¡ "raillery" ¡ "stephen" ¡ "tech". | 0.9844 | 0.9258 |
| Step 3 | So the answer is amperage ¡ crimea ¡ farther ¡ insolent ¡ ping ¡ protocol ¡ raillery ¡ stephen ¡ tech. | 0.9961 | 0.7305 |
| Solution-level Score | | 0.5938 | 0.1641 |
| **Solution selected by FOVER-PRMs (wrong final answer)** | | | |
| Step 0 | First, we look at the first letters of each word. "stephen": "s" (19th letter). "amperage": "a" (1st letter). "protocol": "p" (16th letter). "crimea": "c" (3rd letter). "farther": "f" (6th letter). "raillery": "r" (18th letter). "tech": "t" (20th letter). "insolent": "i" (9th letter). "ping": "p" (16th letter). We now have: ["amperage"] ¡ "crimea" ¡ "farther" ¡ ["protocol" "ping"] ¡ "insolent" ¡ "raillery" ¡ "stephen" ¡ "tech". | 0.5625 | 0.1826 |
| Step 1 | Now let's sort this subpart ["protocol" "ping"] by looking at their second letters. The second letter: "protocol": "r" (18th letter). "ping": "i" (9th letter). We now have: "amperage" ¡ "crimea" ¡ "farther" ¡ "ping" ¡ "protocol" ¡ "insolent" ¡ "raillery" ¡ "stephen" ¡ "tech". | 0.9961 | 0.7305 |
| Step 2 | Now let's sort this subpart ["ping" "protocol"] by looking at their third letters. The third letter: "ping": "n" (14th letter). "protocol": "o" (15th letter). We now have: "amperage" ¡ "crimea" ¡ "farther" ¡ "ping" ¡ "protocol" ¡ "insolent" ¡ "raillery" ¡ "stephen" ¡ "tech". | 0.9727 | 0.9844 |
| Step 3 | So the answer is amperage ¡ crimea ¡ farther ¡ ping ¡ protocol ¡ insolent ¡ raillery ¡ stephen ¡ tech. | 0.9922 | 0.8672 |
| Solution-level Score | | 0.5625 | 0.1826 |

⊗ **Both PRMs assign incorrect step-level scores, with the baseline selecting the correct answer by chance.** The following example shows a response generated from Llama 3.1 8B on GSM8K. In this case, both PRMs assign high scores to both solutions, indicating that neither effectively distinguishes correctness. The baseline PRM happens to select the better solution, but this is due to chance rather than meaningful score differences. Since both PRMs are unsuccessful, this case does not show the degradation by FOVER.

---

**Problem:**

Mark is a copy-editor. He edits an equal number of sentences each week for two different publishers, who each pay him a different rate per sentence. Publisher B pays Mark twice what Publisher A pays. Mark edits a total number of 1000 sentences each week, and Publisher A pays him 5 cents per sentence. How much does Mark make in a week, in cents?

**Ground truth:** 7500

---

| Step | Explanation | Baseline PRM | FOVER-PRM |
|------|-------------|--------------|-----------|
| **Solution selected by the baseline PRM (correct final answer)** | | | |
| Step 0 | Mark edits a total of 1000 sentences each week. | 0.9922 | 1.0000 |
| Step 1 | He edits an equal number of sentences for both Publisher A and Publisher B, so he edits 1000 / 2 = 500 sentences for Publisher A and 500 sentences for Publisher B. | 1.0000 | 0.9258 |
| Step 2 | Publisher A pays 5 cents per sentence, so Mark earns 500 * 5 = 2500 cents from Publisher A. | 1.0000 | 0.9961 |
| Step 3 | Publisher B pays twice what Publisher A pays, so Publisher B pays 2 * 5 = 10 cents per sentence. Mark earns 500 * 10 = 5000 cents from Publisher B. | 1.0000 | 0.9922 |
| Step 4 | In total, Mark earns 2500 + 5000 = 7500 cents per week. | 1.0000 | 0.9922 |
| Step 5 | Therefore, the answer (arabic numerals) is 7500. | 1.0000 | 0.9727 |
| Solution-level Score | | 0.9922 | 0.9258 |
| **Solution selected by FOVER-PRMs (wrong final answer)** | | | |
| Step 0 | Mark edits 1000 sentences each week. | 0.9922 | 1.0000 |
| Step 1 | Publisher A pays him 5 cents per sentence, so Publisher A pays him 1000 x 5 = 5000 cents per week. | 1.0000 | 0.9883 |
| Step 2 | Since Publisher B pays twice what Publisher A pays, Publisher B pays 2 x 5000 = 10,000 cents per week. | 0.9961 | 0.9766 |
| Step 3 | The total amount Mark makes in a week is 5000 + 10,000 = 15,000 cents. | 0.9961 | 0.9453 |
| Step 4 | Therefore, the answer (arabic numerals) is 15000. | 1.0000 | 0.9453 |
| Solution-level Score | | 0.9922 | 0.9453 |

● **FoVer-PRMs assign wrong step-level scores, although the baseline PRMs assign reasonable step-level scores.** The following example shows a response generated by Llama 3.1 8B on GSM8K. In this case, the FoVer-PRM assigns a relatively low score to Step 3 of the first solution, even though the step is correct. It also fails to assign a low score to Step 0 of the second solution, which contains an error. This represents a relatively rare instance in which the FoVer-PRM genuinely degrades step-level verification performance.

---

**Problem:**

Each person in a certain household consumes 0.2 kg of rice every meal. Supposing 5 members of the household eat rice every lunch and dinner, how many weeks will a 42 kg bag of rice last?

**Ground truth:** 3

---

| Step | Explanation | Baseline PRM | FoVer-PRM |
|------|-------------|--------------|-----------|
| **Solution selected by the baseline PRM (correct final answer)** | | | |
| Step 0 | Each person consumes 0.2 kg of rice per meal. There are 2 meals per day (lunch and dinner), so each person consumes 0.2 * 2 = 0.4 kg of rice per day. | 0.9688 | 0.9961 |
| Step 1 | There are 5 members in the household, so the total rice consumption per day is 0.4 * 5 = 2 kg. | 1.0000 | 0.9961 |
| Step 2 | There are 7 days in a week, so the total rice consumption per week is 2 * 7 = 14 kg. | 0.9961 | 1.0000 |
| Step 3 | A 42 kg bag of rice will last for 42 / 14 = 3 weeks. | 0.9922 | 0.8516 |
| Step 4 | Therefore, the answer (arabic numerals) is 3. | 1.0000 | 0.9961 |
| Solution-level Score | | 0.9688 | 0.8516 |
| **Solution selected by FoVer-PRMs (wrong final answer)** | | | |
| Step 0 | Each person consumes 0.2 kg of rice per meal, and there are 3 meals a day (breakfast, lunch, and dinner). | 0.7656 | 0.8945 |
| Step 1 | So, each person consumes 0.2 * 3 = 0.6 kg of rice per day. | 1.0000 | 1.0000 |
| Step 2 | Since there are 5 members, the total rice consumption per day is 0.6 * 5 = 3 kg. | 1.0000 | 0.9961 |
| Step 3 | A 42 kg bag of rice will last for 42 / 3 = 14 days. | 0.9609 | 0.9531 |
| Step 4 | Since there are 7 days in a week, the rice will last 14 / 7 = 2 weeks. | 0.9961 | 0.9805 |
| Step 5 | Therefore, the answer (arabic numerals) is 2. | 1.0000 | 0.9883 |
| Solution-level Score | | 0.7656 | 0.8945 |

## L  LICENSE

We release our dataset under Creative Commons Attribution 4.0 International and our code under Apache License 2.0. Our dataset and code are based on the following resources. We consider our license to be (one-way) compatible with all licenses listed below.

**Datasets.** FOVER-80K is based on the following datasets.

- FLDx2 (Morishita et al., 2024): Creative Commons Attribution 4.0 International [8]
- GSM8K (Cobbe et al., 2021): MIT[9]
- MetaMathQA (Yu et al., 2024): MIT[10]
- Big-Math (Albalak et al., 2025): Apache License 2.0[11]

**Code and packages.** Our code is partially based on the following resources.

- FLD (Morishita et al., 2024): Apache License 2.0[12]
- Isabelle: BSD-style regulations[13]
- Neural theorem proving tutorial (Welleck, 2023): MIT[14]
- DTV (Zhou et al., 2024): MIT[15]

## M  COMPUTATIONAL RESOURCES AND EXECUTION TIME

**LLM training and inference.** We use four NVIDIA A100 SXM4 80GB GPUs for training and inference. Training each 8B-class model on our dataset takes approximately one hour using our training data, FOVER-80K. Evaluation requires considerably more time because of the Best-of-K evaluation, and reproducing all the evaluation in this paper will take approximately three days.

**Dataset Creation.** Dataset creation in the formal logic tasks is efficient, and the verification for our dataset takes a few hours. However, verification of formal proofs using Isabelle is CPU-intensive and takes a long time. We run 40 parallel Isabelle processes across four servers, verifying approximately 200 proofs per hour. The servers have different CPUs, with the most powerful one equipped with two AMD EPYC 7763 64-Core Processors. The verification of our formal proof dataset takes about two weeks in total. We can accelerate the verification by running more Isabelle processes in parallel.

---

[8] https://github.com/hitachi-nlp/FLD-corpus/blob/neurips_2025/LICENSE
[9] https://github.com/openai/grade-school-math/blob/master/LICENSE
[10] https://huggingface.co/datasets/meta-math/MetaMathQA/blob/main/README.md
[11] https://huggingface.co/datasets/SynthLabsAI/Big-Math-RL-Verified/blob/main/README.md
[12] https://github.com/hitachi-nlp/FLD/blob/neurips_2025/LICENSE
[13] https://isabelle.in.tum.de/
[14] https://github.com/wellecks/ntptutorial/blob/main/LICENSE
[15] https://github.com/jinpz/dtv/blob/main/LICENSE

