# OpenReview forum: "Generalizable Process Reward Models via Formally Verified Training Data"
_ICLR.cc/2026/Conference — ICLR 2026 Conference Withdrawn Submission_

### Official Review · Reviewer_uVpU · 2025-10-29

**Soundness:** 2
**Presentation:** 3
**Contribution:** 2
**Rating:** 4
**Confidence:** 4

**Summary:**

This work proposes FOVER, an approach to synthesize PRM training data with accurate step-level error labels automatically annotated by formal verification tools. The PRM trained on the synthesized FOVER-80K dataset outperforms PRMs based on the original LLMs and achieves competitive or superior results compared to state-of-the-art PRMs.

**Strengths:**

1. This work addresses a significant problem: the automated synthesis of training data for process reward models (PRMs). The proposed method appears easy to reproduce, and the paper is clearly written. Consequently, the proposed method, dataset, and model can serve as a valuable reference for the community.
2. The experimental finding that data synthesized using formal verification generalizes to other general-purpose tasks is particularly interesting. This result may offer a new perspective on training methodologies for PRMs.
3. The paper presents extensive comparative experiments that validate the effectiveness of the proposed approach.

**Weaknesses:**

1. First, the method used to compare the different PRMs in this paper may not be fair. Given that FOVER-80K is trained on data with formal verification labels, it is expected that the resulting PRM performs well on logic-oriented tasks. Conversely, other PRMs are primarily trained on MATH reasoning, leading to better performance on mathematical benchmarks reasonable as well. Therefore, to assess the models' generalization capabilities, calculating the average performance after excluding the logic and math benchmarks might be more informative. However, the experiments on Llama3.1-8B seem to indicate that the performance differences among PRMs on out-of-distribution (OOD) benchmarks are not significant.
2. The main body of the paper does not mention the model used to convert responses into formal proofs. The authors state in the appendix that for formal theorem proving tasks, FOVER utilizes a more powerful LLM (Llama 3.3 70B) to translate informal solutions into formal proofs. This information should be provided in the main text. Furthermore, the example in Figure 4 shows a GSM8K-level problem. While this choice might be reasonable for small-model scenarios, the same formalization step might become infeasible if the method is extended to more challenging mathematical tasks like MATH and AIME.
3. Following up on the previous point, the authors need to provide more discussion and ablation studies on the data composition. It is clear that the mathematical data component contributes to the performance gains on math benchmarks. It is therefore necessary to discuss whether the performance gains on other benchmarks originate specifically from the Formal Logic data. Formal proof data converted from GSM8K-level problems is unlikely to contain complex logical reasoning; consequently, this portion of the data more likely boosts mathematical capabilities more than logical reasoning. To clarify the specific contributions of **Formal Theorem Proving**, presenting results from models trained on the two data types separately appears to be a necessary experiment.
4. Given its data composition, directly comparing a PRM trained on FOVER-80K with other math-focused PRMs seems unfair. The experiment in Table 5, which involves mixing the datasets, appears to be a more appropriate setup. FOVER should perhaps be viewed as a scalable method for initializing reward model preferences, rather than as a direct method for task-specific comparison. Works such as [1] have proposed methods for preference initialization. Positioning FOVER in this light might help clarify the paper's core contribution.

[1] Dou S, Liu S, Yang Y, et al. Pre-trained policy discriminators are general reward models[J]. arXiv preprint arXiv:2507.05197, 2025.

**Questions:**

First, the PRM trained on FOVER-80K demonstrates notable generalization to other domains, which is an interesting finding. Consequently, if FOVER proves to be a scalable method at a manageable cost, it could represent a viable approach for preference initialization in PRM training. However, the paper's direct comparison between the FOVER-trained PRM and other PRMs is somewhat confusing. The authors should perhaps reconsider the framing of their contribution to better clarify FOVER's specific value to the community.

1. An interesing phenomenon observed is that the PRM based on Qwen2.5-7B exhibits stronger generalization to other domains compared to the one based on Llama3.1-8B. A discussion of this observation will make the experimental conclusions more solid.
2. Could the authors provide an ablation study to validate the respective roles (or contributions) of the two distinct data types within the FOVER dataset?

For other questions, please refer to the weaknesses section.

---

### Official Review · Reviewer_DQSh · 2025-10-29

**Soundness:** 3
**Presentation:** 4
**Contribution:** 2
**Rating:** 4
**Confidence:** 3

**Summary:**

The paper proposes to build process reward models (PRMs) using formal verification tools to ensure that each step in the training data is a valid inference in formal logic or theorem proving. Evaluations are performed on a range of tasks that go beyond the training domains. The paper reports modest improvements in best-of-K over state-of-the-art PRMs such as qwen2.5-math-prm-7b, and mixed results on ProcessBench. The approach appears to be complementary with standard PRM training (PRM800K), as shown in table 5.

**Strengths:**

* Formal verification generalizes the ideas behind RLVR, offering a new source of verifiability that could potentially improve the quality of training data for PRMs.
* The generalization to non-formal tasks like BBH and MMLU is particularly promising.
* The paper is particularly clear in its description of the method and the experiments.
* The paper is accompanied by a dataset release of formally-verified reasoning steps.

**Weaknesses:**

* As noted in the summary, while the approach is interesting the results are mixed at best.
* Conceptually, it is not clear that formal validity alone is enough for a PRM. The derivation steps should also lead to a solution to the problem. It is easy to imagine hacking a PRM based on FOVER by producing a series of steps that are valid but pointless.
* Relatedly, it would be easier to evaluate the utility of the FOVER PRM with best-of-K for larger values of K (e.g., sweeping over 5, 10, 25, as in Lightman et al)

**Questions:**

- Would it be possible to evaluate FOVER PRM at larger values of K?
- How might one combine formal verification with some notion of goal directedness?
- Would it make sense to use formal verification instead as a post-training technique, teaching the model to generate utterances that are valid?

---

### Official Review · Reviewer_DiVu · 2025-10-31

**Soundness:** 2
**Presentation:** 3
**Contribution:** 2
**Rating:** 4
**Confidence:** 4

**Summary:**

This paper proposes FoVER, a framework for synthesizing training data for Process Reward Models (PRMs) using automatic, step-level error annotations obtained via formal verification tools such as Z3 and Isabelle. The central goal is to circumvent the labor-intensive need for human annotation in PRM data creation and enable PRMs to generalize beyond mathematical reasoning to a broad range of reasoning tasks. FoVER synthesizes the FoVER-80K dataset, comprising annotated solutions for logic and theorem proving tasks, and demonstrates that PRMs trained with this data outperform baseline and often rival state-of-the-art PRMs across 12 reasoning benchmarks (including math, logic, NLI, and more). Comprehensive experiments show robust cross-task generalization and the potential of symbolic-verification-driven data to scale PRM training efficiently.

**Strengths:**

1. FoVER creatively applies formal verification tools (Z3, Isabelle) to annotate step-level correctness in LLM-generated reasoning without human supervision (see Figure 3 and Figure 5). This is significant for both efficiency and label quality, as highlighted in Table 1’s comparison with prior PRM work.
2. Unlike prior PRMs restricted predominantly to math reasoning, FoVER shows PRMs trained with its synthetic logic/theorem data can generalize to a wide array of tasks (see Figure 1 for an illustration of the cross-domain deployment). The core result is that 'symbolic-to-informal' generalization is significant for process-based supervision.
3. The paper provides detailed experiments across 12 benchmarks and both Best-of-K selection and step-level error classification. Table 3 and Table 4 present comprehensive, competitive performance, with FoVER-based PRMs typically outperforming baseline LLMs and matching (or surpassing) several recent strong PRMs on out-of-domain tasks, especially in logical reasoning and NLI.

**Weaknesses:**

1. Domain dependence of verification: The FoVER data generation pipeline currently relies on LLM-produced reasoning chains that can be rendered in syntactically valid, formal formats (see Table 2 and formal logic/theorem proving setups). This limits applicability to domains where such formalization is feasible. No experimental evidence is presented for cases where step-by-step formalization is challenging (e.g., commonsense reasoning, legal argumentation, or ambiguous, real-world tasks). This risks overstating the universality of ‘broad reasoning’ generalization.
2. Intermediate step supervision assumes correctness of prior steps: As stated in Section 3.1 and further discussed in Appendix D, step verification is performed under the assumption that all previous steps are correct. For certain reasoning processes, this assumption can lead to undetected compositional errors, where a cascade of mistakes is overlooked. This limitation is not fully surfaced in the main text, and no end-to-end joint verification alternative is discussed.
3. Error labels may not reflect true “real-world” ambiguity: The use of formal verification implies a binary (correct/incorrect) distinction at each step, which may fail to capture the spectrum of plausibility in informal text-based reasoning. The current approach may thus yield PRMs that are sub-optimal for tasks requiring tolerance for partially correct or imprecise intermediate steps. No analysis is provided of how label granularity or verification strictness impacts generalization or usability.
4. Performance lag in core math benchmarks: While FoVER-based PRMs excel on out-of-distribution and logic/NLI tasks, they can trail state-of-the-art PRMs that are heavily optimized for math, especially on MATH and Olympiad-Bench (see Table 3 and Table 4). For instance, while FoVER-PRMs lift Llama 3.1 (8B) step-level AUROC on MATH and Olympiad (74.1/74.8) over baseline (68.8/67.3), models trained on massive math-specific human annotations (cf. Qwen2.5-Math-PRM-7B) can achieve substantially higher (95.3/94.8). This suggests synthetic symbolic data, though generalizable, may have ceiling effects in highly specialized tasks.
5. Potential for “domain leakage”: The use of stronger LLMs for informal-to-formal conversion in theorem proving (see Appendix G.2, Figure 5) could introduce artifacts or over-regularization patterns into the FoVER-80K dataset. There is no discussion of safeguards against such leakage or its possible impact on downstream PRM behavior.

**Questions:**

1. How does FoVER perform when the source reasoning tasks are less amenable to formalization—e.g., commonsense or open-world dialogue? Is there empirical evidence for or against meaningful cross-domain generalization in such settings, or is it mainly limited by data constructability?

2. Section 3.1 and Appendix D briefly mention that verification is performed under a “preceding steps correct” assumption. Did you consider joint, end-to-end step sequence verification, and what practical issues prevented this? Are there known cases in your data where errors slip through due to this assumption?

3.  Is there a systematic characterization of tasks where FoVER-trained PRMs fail or produce counterintuitive scores? Can you share confusion matrices or breakdowns by task/category?

4. Is there a chance that models trained heavily on logic and theorem-verification data pick up artifacts specific to formal reasoning and underperform on “human-judgment” steps (where justification, plausibility, or partial credit matter)?

---

### Author Response · Authors · 2025-11-15

We sincerely appreciate the time and effort you dedicated to reviewing our paper. After careful consideration, we have decided to withdraw the submission. Thank you again for your thoughtful feedback.

---

### Note · Authors · 2025-11-15

I have read and agree with the venue's withdrawal policy on behalf of myself and my co-authors.